# Community-level signatures of ecological succession in natural bacterial communities

Alberto Pascual-García [1,2] & Thomas Bell [1]

A central goal in microbial ecology is to simplify the extraordinary biodiversity that inhabits natural environments into ecologically coherent units. We profiled (16S rRNA sequencing) > 700 semi-aquatic bacterial communities while measuring their functional capacity when grown in laboratory conditions. This approach allowed us to investigate the relationship between composition and function excluding confounding environmental factors. Simulated data allowed us to reject the hypothesis that stochastic processes were responsible for community assembly, suggesting that niche effects prevailed. Consistent with this idea we identified six distinct community classes that contained samples collected from distant locations. Structural equation models showed there was a functional signature associated with each community class. We obtained a more mechanistic understanding of the classes using metagenomic predictions (PiCRUST). This approach allowed us to show that the classes contained distinct genetic repertoires reflecting community-level ecological strategies. The ecological strategies resemble the classical distinction between r- and K-strategists, suggesting that bacterial community assembly may be explained by simple ecological mechanisms.

---

[1] Department of Life Sciences. Silwood Park Campus, Imperial College London, Ascot, UK. [2] Present address: Institute of Integrative Biology, ETH-Zürich, Zürich, Switzerland. ✉email: alberto.pascual.garcia@gmail.com

The microbial communities inhabiting natural environments are unmanageably complex. It is therefore difficult to establish causal relationships between community composition, environmental conditions and ecosystem functions (such as rates of biogeochemical cycles) because of the large number of factors influencing these relationships. There is great interest in developing methods that reduce this complexity in order to understand whether there are predictable changes in community composition across space and time, and whether those differences alter microbe-associated ecosystem functioning. The most common approach has been to search for physical (e.g., disturbance) and chemical (e.g., pH) features that correlate with community structure and function. This approach has often been successful in identifying some major differences among bacterial communities associated with different habitats[1] and some of the edaphic correlates[2]. However, even if significant correlations between environmental variables and microbial functioning are found, we are still far from understanding the underlying biological mechanisms explaining these relationships. For instance, adding variables such as biomass or diversity to models in which environmental variables are good predictors of function do not strongly improve model predictions[3], suggesting that there is a need for variables that increase the accuracy of biological processes[4].

The development of a more mechanistic picture is hindered for several reasons, such as difficulties in identifying the relative role of stochastic and deterministic processes in shaping microbial communities[5–7], and the pervasiveness of functional redundancy[8,9] and of priority effects[10]. In addition, it is often difficult to identify which functions to assess. Microbes inhabitating a host sometimes have a substantial impact on host performance, for example, turning a healthy into a diseased host[11]. Such extreme impacts of individual taxa make it relatively simple to infer a direct link between community composition and function. In open, natural environments (e.g., soil, lakes, oceans), the impact of individual taxa on ecosystem functions is often minor and generalisations may depend on subjective choices of which functions to measure.

An important step forward comes from manipulative experiments in natural environments, which have identified variables such as pH[12], salinity[8], sources of energy[13], the number of species[14] and environmental complexity[4] as key players in the relationship between bacterial community structure and functioning. Improved control can be obtained by domesticating communities surveyed from natural environments by growing them in a synthetic (albeit complex) environment, and quantifying their functioning under such controlled conditions[15,16]. With these experiments, it becomes possible to directly test the hypothesis that more similar communities have more similar functions without the confounding influence of extrinsic environmental conditions.

Community similarity can be assessed using a rich array of analytic tools that identify β-diversity clusters within multivariate data sets, such as the detection of communities in species co-occurrences networks[17] or the reduction of the dimensionality of β-diversity similarities[18]. These approaches have been pervasive in the medical microbiome literature, for example, in the search for enterotypes—i.e., whether individuals are characterised by diagnostic sets of species representing alternative community states[18,19] which, in this paper, we call "classes" of communities. The existence of classes in communities sampled from different locations may be due to variable environmental conditions that select for different taxa, or may be explained more parsimoniously by stochastic processes together with strong dispersal limitation[20]. Deciphering the likelihood of different ecological mechanisms can be assessed by adopting a suitable null model,

for example, see ref. [21]. Community classes arising from environmental selection would also be functionally different, whereas we would not expect functioning to differ among community classes created by stochastic processes.

Once classes and functional differences have been identified, it is possible to step down into key biological processes by focusing on the genetic repertoires of the constituent taxa[22]. Investigating the dominant genes present in the different community classes allows explanations of functional differences and the determination of ecological strategies. For example, community classes that differ in genes related with environmental sensing, degradation of extracellular substrates, or metabolic preferences, could be used as hypotheses of the molecular mechanisms responsible for functional differences. Therefore, the last step aims to explain how the functional and genetic differences arise from the prevailing environmental conditions[23], and could point to the specific environmental parameters that could be measured. This approach solves the problem of measuring many environmental parameters in the hope that some will be significantly associated with community structure or ecosystem functioning. Lack of any clear functional differentiation among community classes is also informative, and would indicate alternative community states with redundant functions[24–26]. Such redundancy could arise in the absence of environmental variability, which could also help explain the lack of a dominant environmental axis that explains variation in composition.

In this work, we followed the above pipeline using a large data set consisting of >700 samples of rainwater-filled puddles (phytotelmata) that can form at the base of beech trees. The bacterial communities present in the tree-holes are key players in the decomposition of leaf litter, and therefore of great interest more broadly for understanding decomposition in forest soils and riparian zones. This is an ideal system to follow the above pipeline given the relatively similar conditions found across different locations, making it unique in terms of replicability of a natural aquatic environment[27,28], and its relatively low diversity. Indeed, although effects caused by environmental variation on phytotelmata ecosystems have been investigated in meio- and macrofaunal communities[28], the influence in microbial communities is largely unknown. Moreover, prior work has emphasised bottom–up drivers of tree-hole diversity like nutrients[29–31], but top–down approaches that may help us understand other drivers of microbial composition like stochastic dispersion or interactions have received less attention[28].

Previous work using this data set showed that rare taxa influenced narrow functions (degradation of specific substrates), whereas abundant taxa influenced broad functions (overall community productivity)[32]. Here, we aim to illuminate the mechanistic basis of this relationship. The large data set allow us to study natural variation in bacterial community composition through the top–down categorisation of communities into classes. We then link the classes with bacterial functioning, analysing a set of community-level functional profiles obtained from laboratory assays of the same communities[32], and investigating whether the classes differed in their functional capacity. Instead of focusing on each function individually, we investigate how the functional profiles varied across the community classes. We then use metagenomic information to understand whether similar compositions and functions are translated into different classes of genetic repertoires.

We find significant differences in the genetic repertoires and functional measurements among classes, which we interpret in the context of changing environmental conditions. We address whether differences in the communities are owing to the historical processes at the different geographic locations, or if they are rather more influenced by contingent local conditions. These

factors are often difficult to resolve[26] but may both be important owing to the high temporal variability in these systems, as observed in compost ecosystems[33]. Interestingly, interpreting the signatures found in the functional measurements and in the genetic repertoires lead us to hypothesise the existence of community-level ecological strategies, reflecting an ecological succession driven by local environmental dynamics of the tree-holes. These ecological strategies resemble the classical distinction between r– and K–strategists described for single species[34].

## Results

**Microbial community classes are determined by local conditions**. We analysed 753 bacterial communities sampled from water-filled beech tree-holes in the southwest of the UK[32] (see Supplementary Table 1 and Supplementary Fig. 1). Communities were grown in a medium made of beech leaves as substrate for 7 days and then their composition interrogated through 16S rRNA sequencing (see Methods). We analysed the $\beta$-diversity of these communities according to two different metrics: the Jensen-Shannon divergence ($D_{JSD}$)[35], and a transformation of the SparCC metric ($D_{SparCC}$, see Methods[36]).

We found that there was a strong relationship between spatial distances and the two $\beta$-diversity distances (Mantel test: $r = 0.21$; $p < 10^{-3}$ for $D_{SparCC}$ and $r = 0.19$; $p < 10^{-3}$ for $D_{JSD}$). This correlation was unexpected because the communities were sequenced following cryo-preservation and subsequent growth under laboratory conditions, so the communities did not necessarily reflect their composition in the original environments. To test if this trend was maintained across the different scales, we clustered samples that were closer in space, and retrieved the classifications found at 10 distance thresholds spanning five orders of magnitude (from $< 5\,\text{m}$ to $>100\,\text{km}$). We used three statistics (ANOSIM, MRPP and PERMANOVA[37,38], see Methods) to test whether the $\beta$-diversity distances within clusters were significantly smaller than those between clusters for the 10 classifications. In all cases, the three tests supported the hypothesis that communities within locations were significantly more similar than between locations (permutation tests, $p < 10^{-3}$, see Supplementary Fig. 2).

We studied how the statistics changed across the 10 distance thresholds. We observed an increase in the mean community dissimilarities within clusters (quantified with the MRPP statistics) and a decay in the ANOSIM $R$ statistics (Fig. 1c and d), whereas PERMANOVA remained roughly constant across scales (Supplementary Fig. 3). To interpret these trends, we analysed the behaviour of these metrics with synthetic data in which artificial $\beta$-diversity distances matrices were generated under different scenarios that altered the mean and variance in $\beta$-diversity distances within and between locations and across scales (Supplementary Figs. 4–5). The increase in the MRPP statistics with increasing spatial distance in the experimental data may be indicative of an important role for dispersal limitation[20]. However, the distance-decay in the ANOSIM-$R$ statistic matched the experimental data (Fig. 1d) only when the variance of the simulated $\beta$-diversity distances was large (Supplementary Fig. 7 middle column, bottom). To give a sense of the implications of this finding, 3% of the $\beta$-diversity distances between samples 100 km apart should be as high as those within 5 m of each other (Supplementary Fig. 8, right). Such a finding either could indicate substantial long-distance dispersal over 100 km, or alternatively that there are similar selection pressures at some distant locations.

We explored this alternative hypothesis that similar communities found at distant locations result from similar underlying environmental conditions. We performed unsupervised clusterings with $D_{JSD}$ and $D_{SparCC}$, revealing in both cases six distinct

community classes (Fig. 2a and b, Supplementary Figs. 9–10 and Supplementary Table 2 for global characteristic metrics such as diversity). The whole set of communities are dominated by *Proteobacteria*, and the community classes were distinguished at the genus level (95% sequence similarity), including a higher presence of the genera *Klebsiella* and *Pantoea* (classes 1, red; and class 3, pink); *Paenibacillus* and *Sphingobioum* (class 2, green); *Serratia* (class 5 blue); *Sphingomonas*, *Streptomyces* and *Pseudomonas* (classes 4, yellow) and low abundant genera like *Brevundimonas* and *Herbaspirillum* and, again, *Pseudomonas* for class 6 (grey). In the following, we refer to class 1 (red) as the reference class because it encompassed the largest number of communities (Supplementary Table 2). We refer to the remaining communities by their most-distinctive taxon as *Paenibacillus* (class 2), *Klebsiella* (class 3), *Streptomyces* (class 4), *Serratia* (class 5). For class 6, we observed that although the *Pseudomonas* genus was also high in other communities, classes 4 and 6 were dominated specifically by *Pseudomonas putida* (Supplementary Fig. 10), which we selected as representative of class 6. In ref. [39], we use a network approach to identify modules of co-occurring species that confirm the key role of the taxa selected as representatives.

We illustrate how these classes are distributed in space by representing the class identity of each community as a coloured bar, alongside the site and date in which the community was sampled (Fig. 2c). As expected from the previous analysis, some communities belong to the same class even if they were distant in space. This can be noted in the dendrogram of Fig. 2c, which shows how distant sites (dendrogram) could have similar compositions (colour, representing the classes; see also examples in Supplementary Fig. 11). In addition, in Fig. 2c, we have highlighted in the figure (dotted rectangles) some of the cases in which there is a better correspondence with the date of sampling than with the site. To test this observation, we showed that a classification based on the date of sampling is consistently more similar to the $\beta$-diversity classes than the site (Supplementary Table 3). Moreover, computing the ANOSIM statistics when tree-holes are clustered according to sampling location (Site values in Fig. 2e) or according to the sampling date (Day and Month values) consistently showed that the specific date (Day) is more informative than the site. The Day was also more informative than the Month, suggesting that seasonal environmental conditions were not the main drivers of the similarities, but that they were rather owing to daily variation in local conditions. Notably, the value of the ANOSIM statistics when the classification considered are the community classes, reaches the same value than the one found at 50 m (Fig. 1c).

In summary, the classification successfully grouped communities into just six groups, with communties within classes often separated by far >50 m. In addition, the date of sampling was more informative than the sites. Taken together, the results suggest that the classes capture similarities in local environmental conditions even in tree-holes that were spatially separated by considerable distances.

**Community classes reflect different functional performances**. If environmental conditions determine compositional differences in the communities, we expect that these differences are translated into different community functional capacities. We investigated this question analysing data that quantified the functional performance of the communities[32]. The sampled communities were cryo-preserved after sequencing, and later revived in a medium made of beech leaves as substrate. Cells were grown for 7 days while monitoring respiration and, after this period the following measurements were taken: community cell counts, community

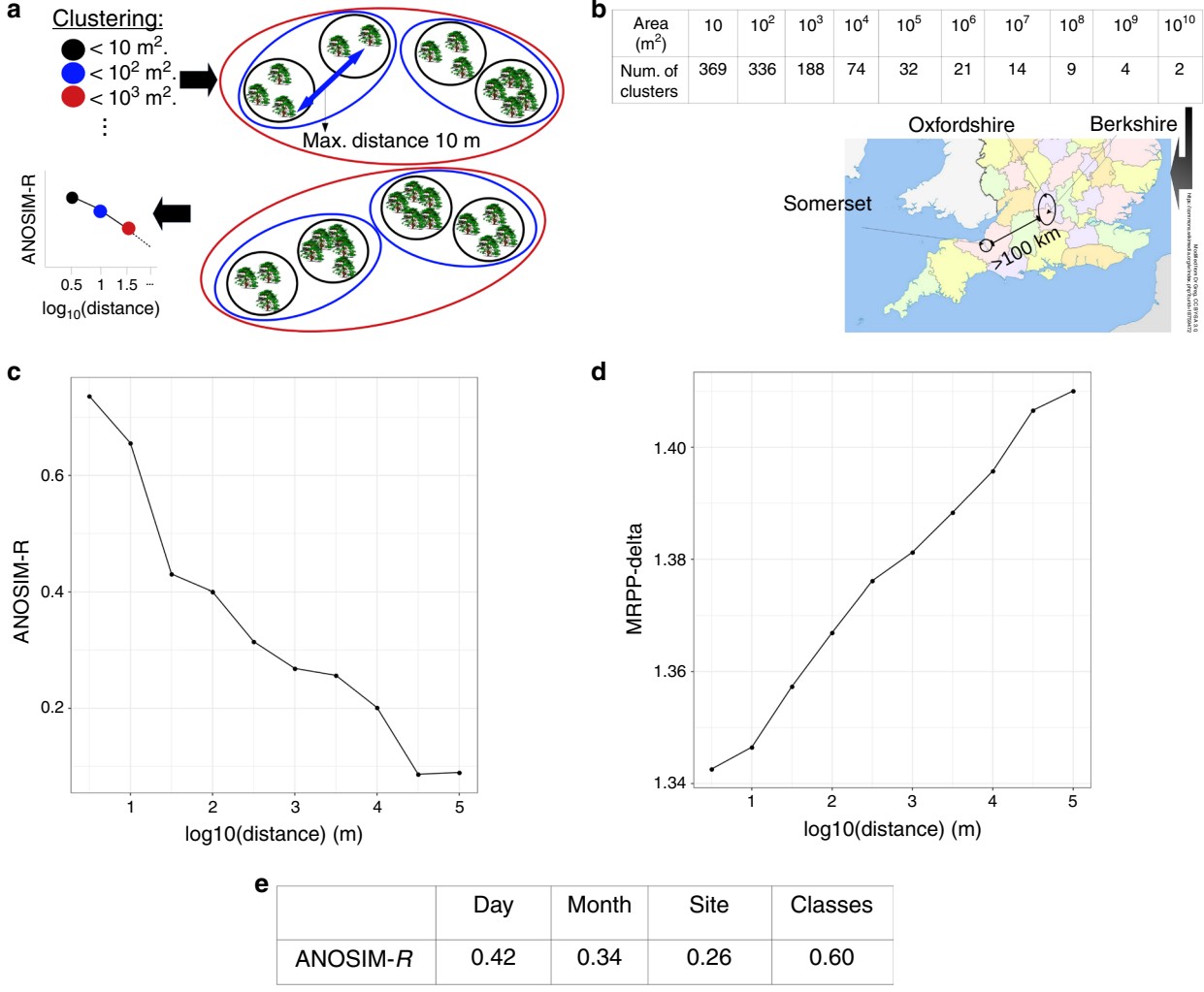

**Fig. 1 Analysis of the decay of community similarity with distance. a** Illustration of the procedure. Trees are clustered within areas $A$ of increasing sizes, leading to one classification every order of magnitude (ellipses of different colours). We approximated these areas by stopping the clustering at spatial distances thresholds equal to $\sqrt{A}$, e.g., the classification for areas of 100 $m^2$ (blue ellipses) is obtained by stopping at $d = 10$ m. For each classification, an ANOSIM and MRPP value was computed to test if the similarity of the communities within the clusters is larger than between clusters, and the statistics were plotted against the spatial threshold. **b** Number of clusters at each spatial threshold. The three counties from which the communities were sampled are shown in the map. The classification at the last threshold consists of two clusters separated by >100 km. **c, d** Observed ANOSIM and MRPP statistics for each of the distance thresholds, reflecting a decay in the similarity of the communities (see Main Text for details). **e** ANOSIM values for communities clustered according to the day and month in which were sampled, the site from which they were collected (optimal classification of spatial distances), and the six community classes (optimal classification of $\beta$-diversity distances).

metabolic capacity (measured as ATP concentration) and community capacity to secrete four ecologically relevant exoenzymes[40] related with (i) uptake of carbon: xylosidase (X) and $\beta$-glucosidase (G); (ii) carbon and nitrogen: $\beta$-chitinase (N); and (iii) phosphate: phosphatase (P).

Visual inspection of the functional measurements shown in Supplementary Fig. 13 indicated substantial differences in the functional capacities among the community classes. In some cases, communities belonging to different classes were clearly separated, which is apparent in the histograms in Supplementary Fig. 13. Therefore, we explored if these differences among the community classes were significant using structural equation models (SEM)[41]. Toward this end, the first step was to identify the most likely structural model that explained how the functions are interrelated. We found a model with an excellent fit (RMSEA < $10^{-3}$, CI = (0–0.023), AIC = 7493 see Methods and Supplementary Fig. 15), showing that measurements related to uptake of nutrients were all exogenous, including ATP

production, cell yields and $CO_2$ production (Fig. 3). In addition, ATP production influenced yield, which in turn influenced $CO_2$. Among exoenzyme variables, N influenced ATP and, notably, only X affected yield, whereas G and P influenced both ATP and $CO_2$.

Assuming that variables are structurally related in the same way independently of the community, we investigated whether the parameters of the model were significantly different for each community class (up to six parameters per pathway, see Methods). To address this question, we considered three scenarios: (i) a model in which all the parameters were constrained to be the same for all the classes; (ii) a model in which each class had a different parameter for each pathway; (iii) an intermediate model, in which some parameters were constrained for some classes. Accounting for penalisations for models with more degrees of freedom (see Methods and Supplementary Results 2), the best model belonged to scenario (iii) (RMSEA < $10^{-3}$, CI = (0–0.035), AIC = 6658, see Fig. 3 and

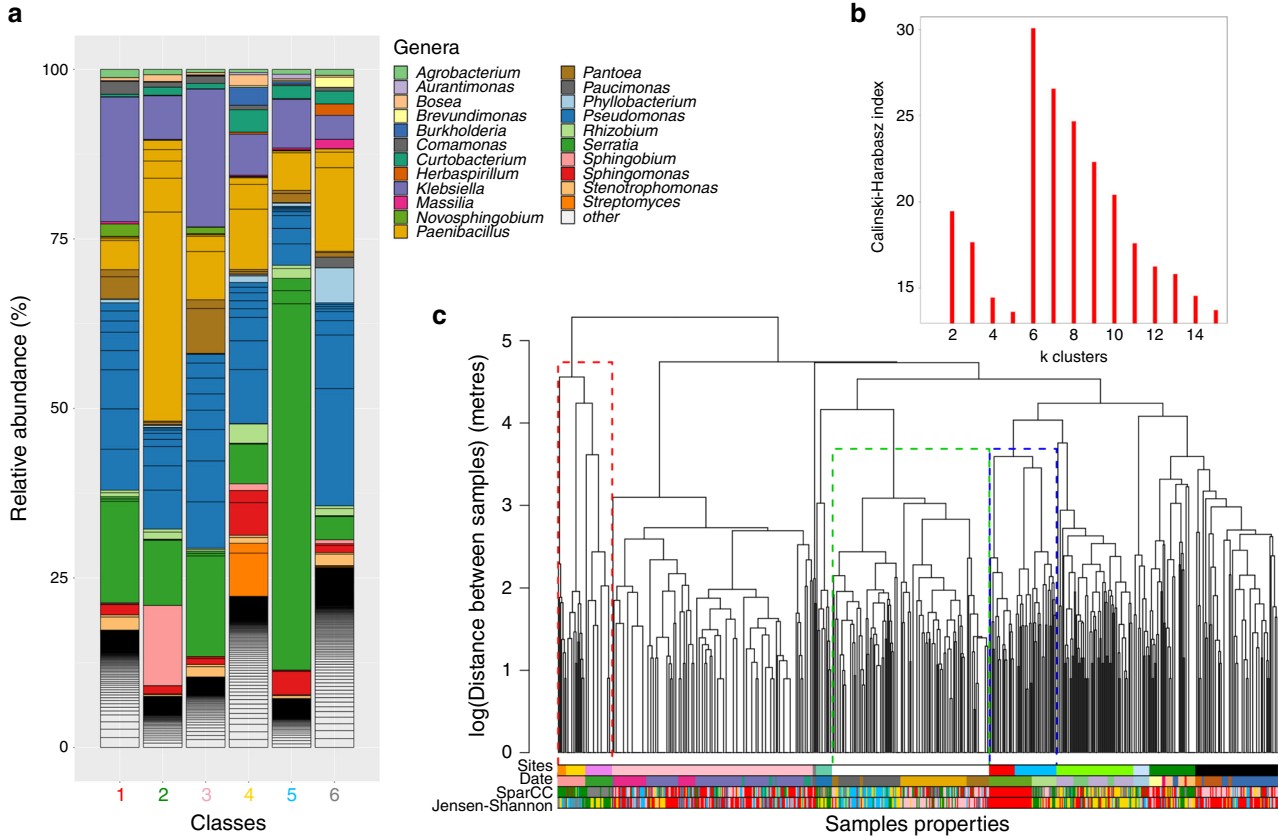

**Fig. 2 Description of community classes and how they relate to the sampling sites. a** Genus frequencies in each of the six community classes found after unsupervised clustering of the communities using $D_{\text{SparCC}}$. Only the 15 most abundant genera in each community class are shown, with the remainder classified as others. **b** Calisnki–Harabasz index versus the number of final clusters in the classification, showing a maximum at six classes. **c** The dendrogram represents the clustering of the samples (leaves in the tree) according to their spatial distances. The coloured bars represent membership of each sample to different classifications (from top to bottom): the sites, the day of sampling collection (Date) and the classes found using either $D_{\text{SparCC}}$ and $D_{\text{JSD}}$. Dotted rectangles in the dendrogram indicate examples of: (green) locations where one site was sampled on different days; (blue) two sites sampled on the same day: (red) a mix of both situations. It is apparent that the date is a better match to the community classes than the site, which is confirmed with a comparison between the classifications (Supplementary Table 3).

Supplementary Table 6). This result supports the hypothesis that the classes had differentiated functional capacities (Supplementary Table 7 and Supplementary Fig. 16).

We then explored whether distinctive pathways for each class could be determined. Given the complexity of the SEM models, we first ruled out the possibility that differences in pathway coefficients were owing to the influence of other (confounding) variables. To control for this possibility, for each pair of endogenous–exogenous variables, we searched for its set of confounding variables with dagitty[42]. Next, for each pair of variables involved in a pathway, we performed a linear regression including its adjustment set of confounding factors, and an interaction term with a factor coding for the different classes. Coefficients should be interpreted as deviations with respect to the reference class (see Methods). The significant interaction terms (Fig. 3d) show how the relationships among the functional variables differed among the community classes. For example, the analysis revealed that cell yield was negatively influenced by β-chitinase activity for the *Paenibacillus* class, for ATP production for the *Serratia* class, while being positive related with β-glucosidase for the classes of *Klebsiella* and *P. putida*. We therefore concluded that the community classes had significantly different functional capacities, which produced the different relationships we observed in the models.

**Community classes depict different genetic repertoires**. To get a more mechanistic understanding of the above results, we analysed the genetic repertoire of each community class by performing metagenomic predictions with PiCRUST[43], and further statistical analysis with STAMP[44]. The Nearest Sequence Taxon Index is 0.059, reflecting a high-quality prediction[43] likely because most of the dominant genera in this system are found in gut microbiomes (e.g., Fig. 5 in ref. [45]).

The fraction of exo-enzymatic genes belonging to *Paenibacillus*, *Streptomyces* and *P. putida* classes was significantly larger than the fractions found for the *Klebsiella*, *Serratia* classes and the reference class, suggesting that the former classes are specialised in degrading a wider array of substrates (Fig. 4).

Clustering the KO annotations into KEGG pathways (see Methods) showed that the 6 community classes differed in their genetic repertoires. Furthermore, these divergent genetic repertoires suggested different ecological adaptations, which are summarised in Fig. 5. Consistent with PCA analysis of the KEGG pathways (Supplementary Figs. 19–22), we divided the classes in two groups: the reference, *Klebsiella* and *Serratia* classes carried the genetic machinery for fast growth, whereas *Paenibacillus*, *Streptomyces* and *P. putida* classes carried the genetic machinery for autonomous amino-acid biosynthesis. Evidence for fast growth in the reference, *Klebsiella* and *Serratia* classes comes from the large fraction of genes related with genetic information

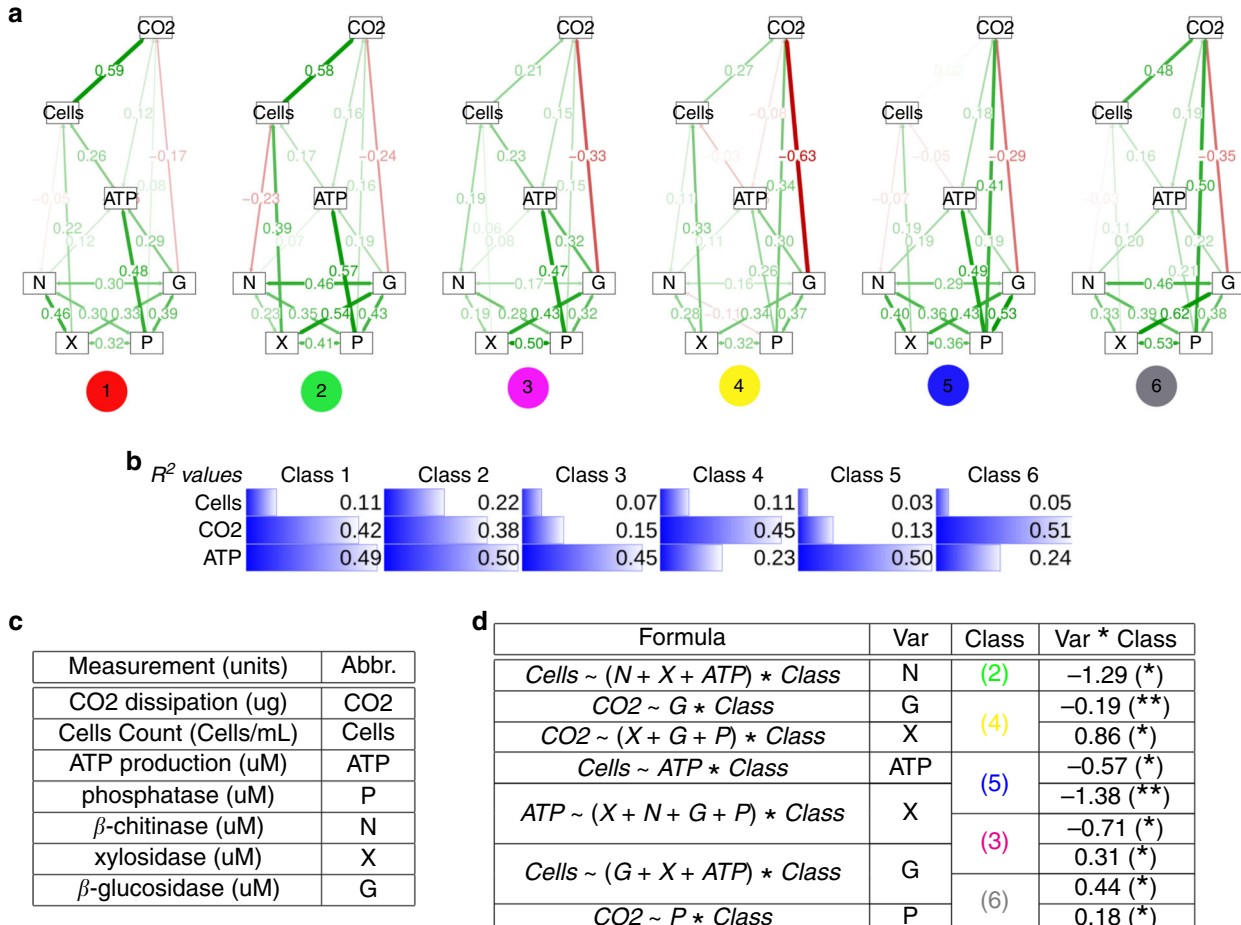

**Fig. 3 Functional capacity of the community classes. a** Diagrams of the final SEMs. The standarized coefficients for each of the six classes are shown on the corresponding pathway. **b** $R^2$ values for the three endogenous variables in the SEM model. **c** Experimental measurements used in this study and their abbreviations. Four independent replicates of each function were measured and then averaged to conduct the analysis. **d** Causal analysis of the influence of the exogenous variable (Var) for each SEM pathway (determined by the response variable and Var in Formula) when the identity of the classes is included as a factor. Confounding factors involved in the pathway are also included (remainder variables in Formula). The column Class reflects the class identity of any significant coefficient found in the interaction terms of the regression with its value shown in the column Var * Class (significance code: *$p$ value < 0.01, **$p$ value < 0.001).

processing (Supplementary Fig. 25), mostly related with DNA replication such as DNA replication proteins genes, transcription factors, mismatch repair, homologous recombination genes or ribosome biogenesis—the latter being a good genetic predictor of fast growth[46]. Second, communities from these classes also carried a larger fraction of genes related with intake of readily available extracellular compounds (Supplementary Fig. 26), including ABC transporters, phosphotransferase system, or peptidases and environmental adaptations including motility proteins, synthesis of siderophores and the two-component systems. Rapid replication often requires a more accurate control of protein folding and trafficking, as the number of proteins and mRNAs increase with increasing growth rates[47]. Consistent with this hypothesis, we found a significantly inflated fraction of genes involved in folding stability, sorting and degradation, including chaperones and genes involved in the phosphorelay system (Supplementary Fig. 27).

A second series of evidences pointing towards orthogonal ecological strategies came from differences in the metabolic pathways associated with the community classes. *Serratia*-dominated class (5) had an inflated fraction of genes related to carbohydrate degradation, including genes involved in glycolysis and in the trycaborxylic acid (TCA) cycle (Supplementary

Fig. 28). In contrast, the *Paenibacillus, Streptomyces* and *P. putida* classes were associated with genes involved in alternative pathways like nitrogen/methane metabolism, and in secondary metabolic pathways related with degradation of xenobiotics/chlorophyl metabolism. Notably, the genes involved in the exoenzymes that were experimentally assayed were higher in these classes, suggesting that they were adapted to environments with more recalcitrant nutrients (Fig. 5). In addition, *Paenibacillus, Streptomyces* and *P. putida* classes had a remarkable repertoire of genes for amino acids biosynthesis–possibly at odds with the reference class and *Klebsiella* and *Serratia* classes, which invested in proteases for amino acid uptake (Supplementary Figs. 29, 30). The apparently low-glycolytic capabilities of these communities could result in pyruvate deficiencies, which would hindered the production of sufficient acetyl-CoA and oxaloacetate required to activate the TCA cycle. Consistent with this observation, we observed that these communities exhibited a significantly larger proportion of genes related with glyoxylate metabolism and degradation of benzoate, which may be used as alternatives to glycolysis (Supplementary Fig. 31). Finally, we observed that communities in the reference class and Klebsiella and *Serratia* classes had a significantly larger repertoire of genes needed to synthesise amino acids requiring pyruvate (valine,

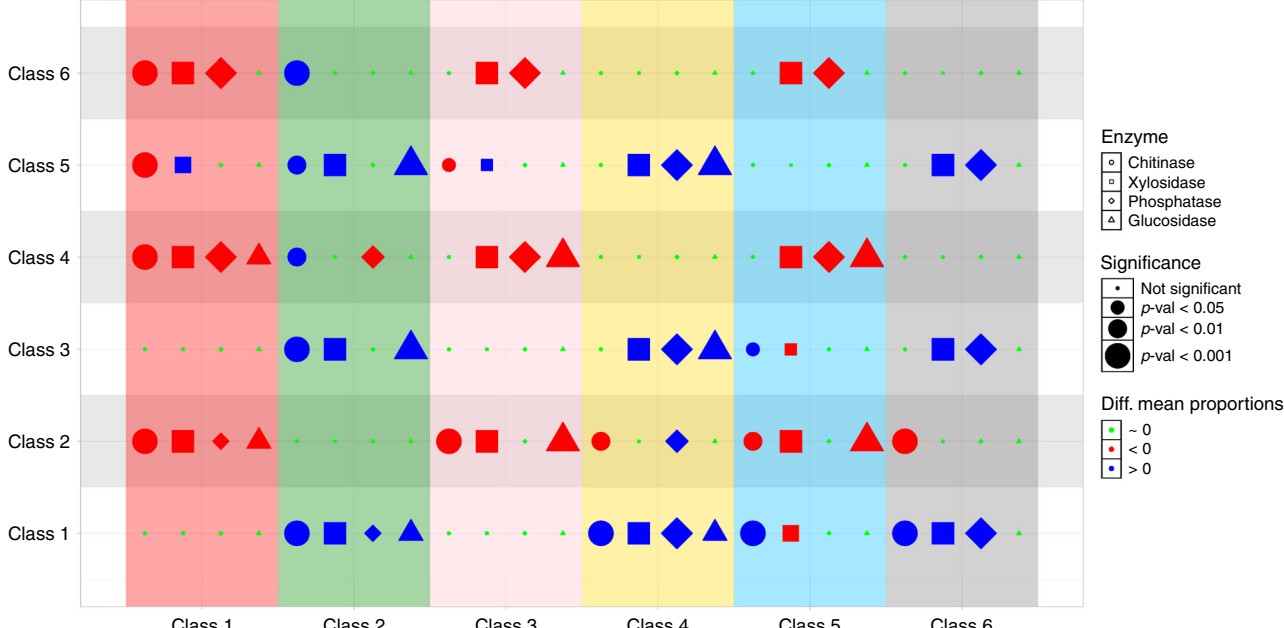

**Fig. 4 Summary of post hoc analysis of exo-enzymatic genes predictions.** The significance of differences in mean proportions of chitinase (circles), β-xylosidase (squares), β-glucosidase (triangles) and phosphatase (diamonds) were tested across all pairwise combinations of community classes. Red (blue) symbols denote that the class in the correspondent column has a significantly lower (higher) mean proportion of the enzyme than the class shown in the row. The size of the symbol is larger for more significant differences. All tests are provided in Supplementary Fig. 4.

leucine and isoleucine), and which, according to our interpretation, they would generate through glycolysis (Supplementary Fig. 28). By contrast, *Paenibacillus*, *Streptomyces* and *P. putida* classes had a significantly larger proportion of genes used to degrade these amino acids (Supplementary Fig. 29) and hence, either they take these essential amino acids from the environment or they generate them from other pathways. Consistent with this observation, genes in these classes were enriched for glycine, serine and threonine metabolism (Supplementary Fig. 29), through which it is possible to obtain valine, leucine and isoleucine, and which could provide an alternative source of acetyl-CoA (Fig. 5).

## Discussion

Our analysis of a large set of tree-hole bacterial communities found a strong distance-decay in the similarity of the communities across several orders of magnitude. The existence of spatial autocorrelation has previously been reported in soil and in other environments[48,12], but this study extends the findings to scales above the short distances (<10 m) previously reported[48]. We suggest that the high ANOSIM statistics we observed require unrealistic levels of dispersal for the pattern to be explained by stochastic processes alone (Supplementary Fig. 29), and therefore points towards a hypothesis that similar environmental conditions occur at distant locations.

We observed that the communities could be arranged into classes, and that the classes corresponded to the site and the date of collection, which are tightly correlated. The finding is consistent with the idea that environmental conditions on a particular day strongly influenced species composition, consistent with previous findings on macro-invertebrate tree-holes communities[49]. Moreover, particular classes were found in different seasons, suggesting that factors like temperature were of secondary importance, despite results highlighting their importance in similar systems[25].

Laboratory experiments confirmed that these classes were associated with different functional capacities, which we believe strongly implies that the classes are ecologically meaningful subgroups. The result was compatible with a scenario of ecological succession in which there was a transition from communities dominated by r-strategists to K-strategists[50]. We suggest that early successional stages were characterised by the *Serratia* class. This class had a negative relationship between ATP and cell yield, indicating low resource use efficiency. In addition, investing in xylosidase had a much lower transfer into ATP production than for the reference class, implying a preference for labile substrates like sugar monomers. Analysis of the metagenome revealed pathways responsible for extracellular degradation and uptake of nutrients, and metabolic processes associated with glycolysis. The class also had many genes associated with environmental processing, fast replication and accurate molecular control of protein folding and trafficking. The mean Shannon diversity of communities belonging to this class was almost the lowest (Supplementary Table 2), which might be expected in a rich environment dominated by a few well adapted fast growers, consistent with the notion of r-strategists.

The next communities in the succession were the reference and *Klebsiella* classes. Although still sharing some of the features of the *Serratia* class, they had distinctive features such as a higher conversion of ATP into yield. Later successional stages were characterised by the *P. putida* and *Streptomyces* classes, exhibiting high respiration values. These classes contained an inflated fraction of genes related to oxidative phosphorylation and were able to synthesise most amino acids. They were also associated with secondary metabolic pathways that may be valuable in environments in which resources are low but where it is possible to scavenge the metabolic by-products of former inhabitants. This is particularly apparent for the *P. putida* class, which also had a higher Shannon diversity, including many rare species, consistent with communities dominated by K-strategists competing for rare and heterogeneous resources.

Finally, the *Paenibacillus* class contained many of the metagenomic characteristics of the *P. putida* and *Streptomyces* classes. It was the class with lowest Shannon diversity, and also a large

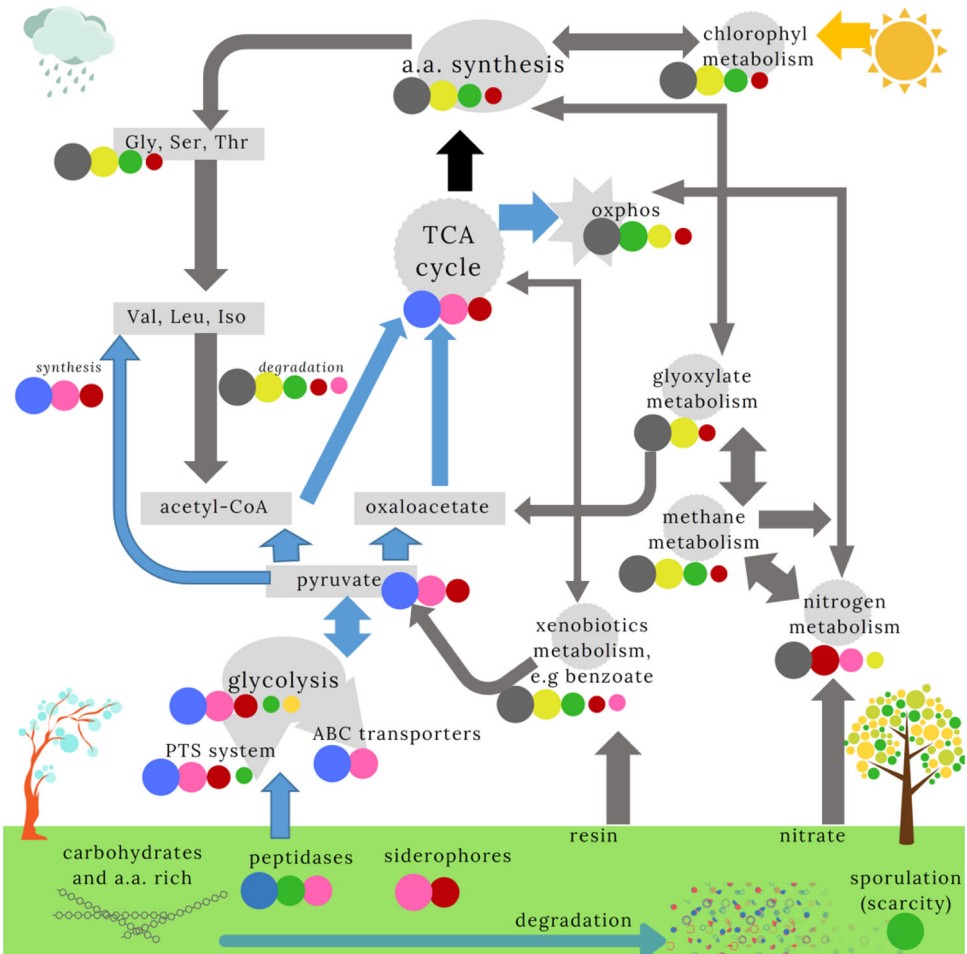

**Fig. 5 Scheme describing the genetic repertoires of the six community classes.** Pathways from the KEGG database that were most relevant for describing the classes are shown. We ranked the mean proportion of genes in each pathway (see Supplementary Results 3), indicated by the size of the circles (Fig. 4). Only classes with significant pairwise post hoc comparisons are shown. *P. putida* (grey) and *Serratia* (blue) classes appear to dominate orthogonal pathways. We therefore indicated how the pathways were influenced by the dominant community (indicated by the arrow colour). The link between the TCA cycle and amino-acid synthesis (black arrow) is unclear. We further illustrate the substrates and hypothetical environmental conditions expected for *Serratia* (rain/cold) and *P. putida* (dry/hot) communities. We suggest the other communities are intermediates between these two classes. The *Paenibacillus* class, with a large number of sporulation and germination genes, may reflect conditions of very low nutrient availability.

fraction of sporulation and germination genes (Supplementary Fig. 32). These results imply that these communities lived particularly unproductive environments. The laboratory results are consistent with this hypothesis: this community had the largest conversion of chitinase activity into yield, which may reflect its ability to take advantage of the remaining nutrients such as dead arthropod exoskeletons or fungi. Water volume is among the main driver of fungi sporulation in this system[51], which would match our interpretation. Taken together, the results imply this class is the last stage of the succession, where nutrients have been depleted to low levels.

There are several environmental conditions that might be driving succession. First, succession may be owing to nutrients dynamics in the tree-holes. A main source of carbon is beech leaf litter, supports meio- and macrofaunal communities[52,53]. Degradation of leaf litter would be compatible with the succession described. Following leaf fall, any imple sugars would rapidly be used over days to weeks, whereas starch and cellulose degrade much more slowly[54]. If this is the main driver of succession in tree-holes, we would expect a strong seasonal signal, with a class dominating in autumn. Our data do not support these observations because the month of the year was a relatively poor classifier

of the samples, and members of the classes we identified were often from different times of year.

Second, succession may be owing to patterns of rainfall. Rainwater can bring nitrogen, sulphate and other ions into the tree-hole, but the pathway followed by the water (stemflow or throughfall) will influence the final chemical compositions[29,31]. For example, flushing after heavy rain can reduce phosphate levels to a minimum[30], and labile orthophosphate is expected to increase at later successional stages[31]. In addition, a progressive acidification in tree-holes that do not receive water inputs for long periods is also expected due to nitrification[29,31]. Rain pulses can therefore have rapid impacts on tree-hole conditions and may explain the similarity of some samples collected at the same date even at distant locations, whereas other properties of the tree-holes like size, litter content and the modes of water collection may preclude complete synchronisation.

We envisage a scenario in which rain events were the primary drivers of bacterial composition, illustrated in bottom-left corner of Fig. 5, which would be modified by tree-hole features (e.g., volume, leaf inputs). Rain would generate pulsed resources of different type and frequency[55], and tree-holes features would determine the rate of resource attenuation[56]. For

instance, large tree-holes or those with large leaf contents would have a slower rate of succession, as resources are depleted less rapidly. This hypothesis would explain why, on some dates, all the tree-holes had similar compositions (recent rain or long standing drought conditions), whereas, beyond that, the classes are distributed across different dates and sites (owing to the differential tempos of succession in tree-holes with different features).

Dissolved oxygen may be a third environmental component that influences community composition. The *P. Putida* classes were associated with genes involved in aerobic respiration and high levels of phosphate. We observed an increase in abundances of strict aerobes, including *Brevundimonas*, *Paucimonas* and *Phyllobacterium*. There was also an increase in genes related with metabolism of nitrate, methane, degradation of benzoate (likely associated with the presence of resines), or chlorophyl (which indicates an increase in photo-heterotrophs). This class might also be able to run the TCA cycle generating acetyl-CoA from acetate, and from the degradation of valine, leucine and iso-leucine, further complemented with glyoxylate metabolism and the degradation of benzoate to generate oxaloacetate. Finally, the class was found in summer and winter, and clustered in specific areas. This makes it less likely that temperature is an important variable, and points towards the amount of water and oxygen as key variables. This observation could also hold for the *Paeniba-cillus* class, for which long drought periods could lead to lack of water regardless of other tree-holes features (Supplementary Fig. 11).

We cannot rule out other site-based conditions like the type of forest management. A study analysing this factor did not find substantial differences in enzymatic activities despite different community compositions[57], perhaps because the low number of samples did not bring sufficient resolution. Another possible local influence for the composition are trophic ecological interactions, like the prevalence of invertebrates in certain areas (e.g., mosquito larvae)[58]. Insects with flying stages may also influence dispersal among tree-holes, which might contribute to microbial commu-nity similarity within a site[59], resulting in a metacommunity structure[49].

The approach taken here provides detailed insights into the community ecology of the bacterial communities inhabiting rainwater pools. By identifying community classes a priori, we were able to piece together the natural history of this envir-onment from the perspective of the bacterial taxa. The spatial and temporal distribution of these classes, combined with the inferred metagenomes, indicate how environmental conditions reflect the metabolic specialisations of the dominant members. In this way, we were able to identify classes resembling r- vs. K-strategists[50] inhabiting tree-holes that were at different suc-cessional stages, a distinction also apparent in gut's micro-biomes[60]. Although this is no doubt an oversimplification, in general we find this conceptual framework is useful for microbes[34], as this ecological dichotomy may well be supported by thermodynamic[61] and protein-allocation trade-offs[62], which might also underlie other observed life history tradeoffs in microbes (e.g., olitgotrophic vs. copiotrophic strategies,[63]). We believe this approach of identifying community classes a priori, therefore, holds great promise for reducing the complexity of microbial community data sets[64], particularly in systems where the microbial communities have not yet been well char-acterised. Combined with the experimental approach of grow-ing the communities under standardised laboratory conditions, the method holds promise for connecting the community classes to distinctive functional properties. In these systems, the approach we have used would generate hypotheses that could become the focus of future experiments or more-detailed

sampling strategies, therefore forming the basis of a bottom–up synthetic ecology that can be predictive in the wild.

## Methods

**Data set.** We analyzed 753 bacterial communities sampled in from rainwater-filled beech tree-holes (Fagus spp.) from different locations in the South West of United Kingdom, see Supplementary Table 1. In total, 95% of the samples were collected between 28 of August and 03 of December 2013, being the remaining 5% collected in April 2014. Spatial distances between samples spanned five orders of magnitude (from < 1 m to > 100 km). Sampled communities were grown in standard laboratory conditions using a tea of beech leaves as a substrate. After 7 days of growth, communities were characterised by sequencing 16S rRNA amplicon libraries from ref. [32]. We considered only samples with > 10K reads, and species with fewer than 100 reads across all samples were removed. This led to a final data set comprising 680 samples and 618 operational taxonomic units at the 97% of 16 rRNA sequence similarity. In previous work[32], four replicates of each of these communities were revived and regrown in the same media, further supplemented with low quantities of four substrates labelled with 4-methylumbelliferon. After 7 days, the experiments quantified the capacity of the communities to degrade xylosidase (abbreviated X in the text, cleaves the labile substrate xylose, a monomer prevalent in hemicellulose), of β-chitinase (N, breaks down chitin, which is the major component of arthropod exoskeletons and fungal cell walls), β-glucosidase (G, break down cellulose, the structural component of plants) and P (breaks down organic monoesters for the mineralisation and acquisition of phosphorus). Cells were also counted at the end of the experiment and $CO_2$ dissipation quantified as a single accumulative measure along the seven days of experiment. Full experimental details can be found in ref. [32].

The rationale for sequencing the communities following growth in the laboratory is that we were primarily interested in the relationship between structure and function. Finding causal relationships between structure and function is made possible here by ensuring that each community is placed in exactly the same environment, as explained in ref. [32]. However, the drawback is that by placing all the communities in a standardised environment, the compositions may not reflect their original composition. We expect community compositions to converge following growth in the standard laboratory conditions, thus any compositional differences that we observed are therefore likely to be conservative estimates of the true differences in the natural communities.

**Determination of classes.** We computed all-against-all communities dissim-ilarities with Jensen-Shannon divergence[35], $D_{JSD}$, and a transformation of the SparCC metric[36], $D_{SparCC}$ (see Supplementary Results 1), and then clustered the samples following a similar approach to the one proposed in ref. [18] to identify enterotypes. In the text, we call these clusters community classes. The method consists of a partition around medoids (PAM) clustering for both metrics, with the function PAM implemented in the R package CLUSTER[65]. This clustering requires as input the number of output clusters desired $k$. We performed the clustering considering a wide range of $k$ values and also computing the Calinski-Harabasz index (CH) that quantifies the quality of the classification, and selecting as optimal classification $k_{opt} = \arg max_k(CH)$, shown in Fig. 2b. Processing of data and taxa summaries provided as Supplementary Results deposited in Zenodo (see Data Availability) were generated with QIIME[66] and Phyloseq[67].

**Community similarity, sampling date and location.** To investigate the relation-ship between the sampling location, the sampling date and the similarity in composition of bacterial communities, we performed analysis of the similarities of the communities grouping them with different criteria and testing if the similarities within groups were significantly different than the similarities between groups, using both $D_{JSD}$ and $D_{SparCC}$. We considered as grouping units one automatic spatial classification and two temporal classifications in which samples are joined in clusters depending on whether they were collected in the same day, or in the same month. Details for the spatial automatic classification and results for two other definitions of sampling sites (see Supplementary Results 1). We clustered the communities in spatial areas $A$ of increasing sizes every order of magnitude, from 10 m² to 100 km², which we approximate considering spatial distances' cutoffs of $\sqrt{A}$ metres. We then computed the ANOSIM, MRPP and PERMANOVA tests (see refs. [37,38]) for each of the resultant classifications, using the R functions ANOSIM, MRPP and ADONIS2, respectively (available in the R package VEGAN[68]) and assessing the significance with permutation tests (10³ permutations). To interpret the observed trends of these metrics we created synthetic distance matrices fol-lowing different criteria, available in Supplementary Results 1.

**Structural equation modelling.** SEM[41] were built and analysed with LAVAAN (version 0.523) and visualised with SEMPLOT R package[69,70]. The modelling procedure was split into different stages detailed in Supplementary Results 2. First, a global model considering all data were investigated following several theoretical assumptions about the relationship between the functions, until a final model was achieved. Then, we looked for a second series of models in which it was possible to fit a different coefficient for each of the parameters in the global model,

constraining the data into subsets corresponding to the community classes (i.e., six possible coefficients for each SEM pathway). Minor re-specification of the model was performed (see Supplementary Results 2). We investigated whether altering the constraints on the models provided better fits, and penalised the models according to the number of degrees of freedom. The main criterion to accept a change was that the Akaike information criterion (AIC) of the modified model was smaller than the original model[71]. We verified that several estimators were improved after any modification, including the RMSEA, the Comparative Fit Index and the Tucker–Lewis Index[72,73].

Investigating causal relationships between endogenous and exogenous variables within the final specified model required controlling for confounding factors. For each pathway in the regression in the SEM model, we identified its adjustment set with dagitty[42]. We then performed a linear regression of each pathway adjusted by the confounding factors, adding a factor coding for the different classes. The coefficients obtained from the regression were estimated with respect to the reference class. Finally, we identified significant interaction terms between classes and the exogenous variable under investigation in the pathway. A significant interaction coefficient involving a given class was interpreted as a different performance of that class with respect to the reference class, and was therefore used to identify distinctive functional features of each class.

**Metagenomic analysis.** Metagenomics predictions were performed using PiCRUST v1.1.2[43] and quality controls computed (Supplementary Table 8). A subset of genes appearing at intermediate frequencies was selected (Supplementary Fig. 18) and aggregated into KEGG pathways[74]. The mean proportion of genes assigned to a specific pathway was computed across communities belonging to the same class. Then we tested if the differences in mean proportions between classes were statistically significant using post hoc tests with STAMP[75] (see Supplementary Results 3). To create Fig. 5 we visually inspected each post hoc test and ranked the classes according with the number of pairwise tests in which they appeared significantly inflated (Supplementary Figs. 23–33). We qualitatively represent this ranking with circles of different sizes. Classes that do not appear inflated in any pairwise test in the pathway are not represented.

**Reporting summary.** Further information on research design is available in the Nature Research Reporting Summary linked to this article.

## Data availability

Original data can be found as Supplementary Material of ref. [15]. A set of processed data used in this work and additional Supplementary Results can be found in Zenodo with the URL: https://zenodo.org/record/3539537.

## Code availability

Code used for some of the analysis presented in the manuscript was deposited in GitHub with the URL: https://github.com/apascualgarcia/TreeHoles_descriptive.git.

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

## Acknowledgements

We acknowledge Damian Rivett for explanations about the experimental methods, and Matt Jones, Lara Durán-Trío and Yonathan Friedman for helpful discussions. We thank Juanjo Abellán and Andreas Steingötter for the support in discussing the statistical methods. The research was funded by a European Research Council starting grant (311399-Redundancy) awarded to T.B. T.B. was also funded by a Royal Society University Research Fellowship. A.P.G. was also funded by the Simons Collaboration: Principles of Microbial Ecosystems (PriME), award number 542381.

## Author contributions

A.P.-G. and T.B. conceived the project. A.P.-G. designed and performed the analysis and wrote the first version of the manuscript. Both authors analysed the data and contributed to the final version of the manuscript.

## Competing interests

The authors declare no competing interests.
