## [Peer review file · Nature Communications]

Reviewers' comments:

Reviewer #1 (Remarks to the Author):

In this manuscript, the authors re-analyze a previously published 16S rRNA dataset collected from microbial communities in beech tree holes (a habitat used by this group for some time now as a model for microbial biogeography). The authors take the unorthodox approach of classifying the communities a-priori based on their community composition and then attempting to characterize the resulting 6 community classes according to the results of different physiological assays (also previously published).

The second part of the paper is dedicated to predicting the communities' metagenomes using PiCRUST. Noting that the main drivers of beta diversity were sampling location and sampling time, the authors use the PiCRUST results to construct a model of functional community succession in beech tree hole microbial communities.

While the analyses were interesting, this manuscript suffers from some weaknesses. The way it is presented is rather disorganized and the text needs through editing for grammar and for simplicity. The methods are not presented in a way that can be reproduced (only a cursory description of software tools that were used it provided). Furthermore, I have some reservations regarding the use and interpretation of PiCRUST here.

Specific comments in order of appearance:

Abstract: edit: Have◇ has

Line 9: edit: major major

Line 10: edit: edaphic

Lines 10-13: the authors provide very little support for this argument. The claims about macroecological results being spurious and difficult to validate are not supported at all by citations. Only the claim about the niche vs neutrality question is supported by a citation but it's not clear to me how this is a problem resulting from the analysis approach, nor how the approaches laid out in this paper will help resolve it.

Line 13: extra comma

Line 13-14: Again, I do not disagree with this argument about "context dependence", but it needs to be supported with citation. Same goes for the argument about "fishing expeditions". Could the authors point to an example of such a fishing expedition?

Line 23: How is this not a fishing expedition? You would still generate "dozens" of measurements to try to identify which explains you're a-priori defined community types.

Line 44: Alternatively, the differentiating environmental parameter was not measured

Line 56: "result in functions to"?

Line 58: edit: analysis ◇ analyses

Line 63: edit: Beech tree

Line 65: the term phytotelmata is introduced here quite suddenly.

Line 67: It is hard to say that "influence on microbial communities is largely unknown" given the large body of work produced by this group on these very communities.

Line 85: "similar compositions and functions were translated into different classes of genetic repertoires, susceptible of a functional interpretation." I do not understand this sentence. What is the different between "functions" and "genetic repertoires" in this context? Is the term "function" used to describe only the results of the physiological/enzymatic assays or also the PiCRUST results?

Line 86: resembling ◇ resembling. In single cells, do the authors mean mono cultures? In any case, a citation is needed here.

Line 87: I do not think "we believe" is an appropriate term for a scientific article. Is the succession an inference? A hypothesis?

Line 90: This would be a good place to refer the reader to table 1 in the supplement and to the unnamed figure below it.

Line 93: This referral to Figure 1 does not give sufficient information to convince the reader that there are indeed 6 classes. First, the authors should also refer the reader here to sup. Fig. 3. Second, just by looking at the figure, I cannot be convinced. The lines connecting the points to the cluster centroids are misleading and distract from the fact that there is substantial overlap in the ordination between the groups (the lines also make this figure difficult to read in general, and I'm not sure what purpose they serve, and the numbers in the centroids are barely legible). The methods state that the PAM method was used to determine these clusters and that six clusters provided the optimal CH index. This step is a critical step for all subsequent analyses. Could the authors provide this data somehow? Plot CH against k as a sup. figure?

Also, when you call the figure in the text, are you calling figure 1a or 1b? please be explicit.

More on Figure 1: should be community classes not "communities classes", genera not "genus", within not "into", throughout, not "alongthrough". Also, genus names should be capitalized (and italicized)

Line 94: communities #1 and #3 are dominated by pseudomonas, but this is not mentioned.

Line 98: This would be a good place to call supplementary figures 1 and 4 (generally, the order of the sup. Figures appears to be rather arbitrary).

Line 105. Class 1 encompasses the largest number of communities. This seems like an important piece of information. In a traditional analyses, the number of samples between different treatments would have been balanced. How balanced are the different clusters? Could the authors present the number of samples within each cluster and their location in an easily readable fashion? A detailed map would be helpful, perhaps added as a panel to Figure 1.

Lines 108-119. I'm familiar with ANOSIM and with distance-decay relationships but I simply could not understand what the authors are doing here. It is not helpful that the methods employed here are partitioned seemingly at random between the Methods and the Supplementary Methods sections, or that no analysis code was provided. How many spatial clusters of samples are included within each cutoff? Is the ANOSIM statistic computed for comparison between spatial clusters within each special cutoff? Why is a high R^2 interpreted as high similarity? Doesn't the log(5) threshold simply include all samples? How is an ANOSIM statistic computed in this case?

Why didn't the authors simply plot pairwise dissimilarity against spatial distance and compute a Mantel test between community dissimilarity and spatial distance matrices?

Line 115: communities'

Line 123: "the specific date turned out to be more informative than the site" Isn't this at odds with the dendrogram in Figure S1? It clearly seems as if Site is the strongest driver of community dissimilarity.

Lines 127-129: Could the authors rephrase this sentence? It is very difficult to read. Also it's not clear what is the part Figure S1 in making this argument.

Line 135: Please edit this text: "there were also measured ATP"

Line 131-135: The authors should make sure and be clear about the fact that these experiments were already published elsewhere. Furthermore, isn't this an example of the very same "fishing expedition" that the authors explain in the introduction that their method is designed to avoid?

Why select these metrics and not others?

Another point that is important to make, is that these aren't measurements of the activities of the communities themselves, but rather of communities that have arisen as a result of cryo-preservation, thawing and growth in a lab-made media. Ref. #18 is somewhat vague about this, but if I understand correctly, the communities were sequenced before, and not after the mesocosm experiment. How can we be sure that the 6 community classes are maintained during/after the mesocosm experiment? Perhaps their phylogenetic makeup completely changed?

Line 138: Supplemental figure 5 seems to appear in the text before sup. Fig 4. In fact, the order of the supplemental figures in the text is as follows: 3,2,1,5,24,4,10-12,15,16,17,18,19-20,21,22.

Sup. Figures 6-9,23,13,14 are not called in the text at all.

Line 138: What do the authors mean when they say that functional capacities were distinct between communities? How is this measured, and how does Sup. Figure 5 show this?

Line 147: "That different coefficients for the classes in some pathways brought a better fit than the model with a single parameter for all classes, supported the hypothesis that the classes had

differentiated function performances." I do not see how this constitutes a valid statistical test. Is this outcome significantly different than an outcome generated by reshuffling the labels (creating sets of 6 classes)?

PiCRUST analysis:

PiCRUST is a type of program that should be used with caution, because of the underlying assumption that the phylogenetic composition is a good predictor of a genetic repertoire. Certainly, broadly this assumption is true, but there are some important caveats and exceptions to this. Importantly, the results depend on the genome database used. Since the genome database is highly skewed towards human-derived bacterial genomes (and indeed this method was designed and benchmarked with human samples in mind), using this method in environmentally-derived samples could be risky, in particular in a rather exotic environment that has contributed a relatively small amount of genomes to the database. This can be potentially circumvented by (i) sequencing and curating a genome database that is specific to this environment or, more realistically, (ii) sequencing metagenomes from a subset of the 753 samples and comparing the PiCRUST predictions to the actual metagenome content. This dataset seems particularly suited to the latter option, and I think adding this data would be of great benefit.

I am not advocating that the authors not use PiCRUST, but rather treat the results with the appropriate amount of caution and attempt to validate the resulting predictions.

It seems like one could roughly classify the classes into two groups: classes 1,3,5 and classes 2,4,6. However, while in Figure 1 it is apparent that classes 1,3 and 4,6 are relatively similar, classes 2 and 5 are outliers, dominated by completely different genera (*Paenibacillus*, *Serratia*). So, the phylogenetic profiles appear to be disconnected from the functional profiles. Furthermore, the authors should consider the compositionality of this data, especially when observing changes in ubiquitous machineries such as the TCA cycle. The results could be affected by differences in predicted mean genome sizes between samples or by differences in species diversity between samples, or by the relative abundance of un-annotated genes.

Reviewer #2 (Remarks to the Author):

Review: Community-level signatures of ecological succession in natural bacterial communities

Here the authors present a compelling dataset of bacterial communities in beech tree holes, which they use as a model to assess bacterial community clusters and ecological succession. Overall the ideas, data and methods are sound. However, I think for background and substance is needed to elevate the reach of the paper. First, community clusters. This is a great example of the use of community clusters, but more content in the introduction is needed to really help guide the field to using this method more often (I also suggest a comparison to traditional diversity measurement, so show how clustering can be more informative). Second, the beech tree hole system is likely low diversity (it is not clear), regardless, better set up why this is such a good model system and how it can help inform assessments in other more diverse (or less) environments. Finally, the ideas around ecological strategies need to be better explained in the introduction so results and discussion are more understandable. Currently the results are quite long and don't do enough to push the main thesis. Overall, an interesting effort and I look forward to the next round.

Abstract:

- 700 samples, give the geographic reference of where/distance covered.
- Define community classes
- Split up the sentence beginning with "We used structural equation modelling..." Also, it doesn't quite make sense since what 'their' is referring to.
- Metagenomic predictions - Based on what?
- Specify the 'ecological succession process' that was observed- which taxa and why.
- Likewise, the last sentence could be more specific
- How can this system inform on other systems? Or at least indicate why this system is a good

model to base understanding of other microbial communities

Introduction

- When classifying communities, presumably the diversity of the system will also effect how clusters are built. It would be good background to add in information on this. Human communities tend to be much more simple than soil communities (for example), so will clustering work equally well on high diverse samples?
- The ability of metagenomic analysis to result in hypotheses is limited by annotation quality – which is still limited. This should be addressed with a reference or 2.
- 'To investigate these questions' be more specific, there were actually a lot of gaps brought up. Same with the following sentence.
- I would like to see come comparison/ contrast hypotheses set up- between what has been found with macro/meso organisms and what is expected with the micro community. This would then make the topic even more relevant – could more strongly bring in biogeography, address if microbes are structured by same processes and macrobes... etc.
- The section about decomposition/function needs a better introduction/background in the previous paragraphs.
- 'we believe'? word choice. Gives evidence maybe.

Results

- The colors should only be listed in the figure legend – in the text explain the results and statistics.
- Where are the statistics to go along with the PCA and how much each axis explains?
- The sentence about uneven sampling and the specific sites. More context is needed, because alone the name of the sites don't mean much.
- Some basic metrics of the samples would be helpful, how many sequences/sample, otus/sample, and put it into context.
- Since the argument that this method will progress the field more, can you compare the cluster results to a typical community diversity Mantel/Permanova. To see if the same variables are relevant, or maybe they are different. But regardless a comparison would help the argument of the paper.
- "To explore ... functional performance...[18]"
- When referring to the clusters – sometimes it is by the number and sometimes by the dominant taxa. I think it should always be by the dominant taxa, this is more informative to the reader and could then be more relevant for future work using similar methods.

DISCUSSION

- Expand reasons for distance decay finding
- There is a lot of speculation about the variability/weak patterns. I would like to see this go beyond just saying 'it could have been this' and really address how this study will help move the field forward. There are interesting analyses that are being overlooked for space on the variability.
- This would also leave more space to really get into hypotheses to link clusters and function.

Reviewer #3 (Remarks to the Author):

Major comments

Authors examined beech tree-holes bacterial communities at large scale and linked six distinct community classes to community-level functional capacity. It is a well-designed top-down approach to link potential bacterial metabolic functions predicted from simple in situ communities to functional performances. It is interesting to see the community types and functions are linked to ecological succession in a predictable way. The study is unique and the manuscript is also well organized. I have only several minor comments below.

Minor comments

There are no line numbers, which makes hard to leave comments line by line.

In this kind of simple communities, phyotypes defined at 97% 16S similarity seem too broad. How about clustering sequences at sub-OTU level such as 99% similarity or exact sequence variants? Around 100% identity threshold for the 16S V4 region was suggested as a proxy for bacterial species (Edgar, 2018).

Another concern is that the total bacterial abundance (or biomass) does not seem to be accounted in this community type vs. function comparison. Is there any information at which orders of magnitude microbial abundance vary among samples? For example, the relatively high proportion of *Paenibacillus* in class 2 may not parallel their absolute abundance. What if their absolute abundance remained at similar level across all samples? Then you still can explain that their dominance is due to the high content of sporulation genes?

In Methods

Dataset

1. Why did you retain >100 reads samples? 100 reads per sample seems too low to be included in any further analysis. Even if you provided in supple that sparCC distance index is not affected by the number of reads, I do not think 100 reads are sufficient enough to reflect real community diversity. Also, those low read numbers are commonly obtained from incomplete PCR reaction or incomplete amplicon sequencing in MiSeq due mainly to low microbial biomass or the presence of PCR inhibitors. Could you provide any evidence of the same or similar results without those too low reads samples?

2. Operational taxonomic units (OTUs)

3. 16S rRNA gene sequence similarity

In Discussion

It is highly speculative that the dominance of *Paenibacillus* in class 2 is ascribed to the large fraction of sporulation and spore germination genes. Different from other class-representative genera, there are not many *Paenibacillus* genomes available, which reduces the accuracy of 16S-based metagenome prediction as discussed below. Is it supported by any evidence obtained from spore counting assay or endospore enrichment experiment? Cells having those genes in their genomes are not necessarily dormant or present in spore forms.

In Supplementary doc

1. The mean weighted NSTI score of 0.059 seems appropriate to use PiCRUST-base gene prediction. However, functional gene prediction results should be carefully interpreted when the number of neighboring genomes varies across major taxa. The number of available neighboring genomes of those representative taxa (*Serratia*, *Klebsiella*, *Pseudomonas*, and *Paenibacillus*) of each community class is different. There are not many genomes available for *Paenibacillus* compared to those of other three genera and in particular *P. borealis* has only one genome available.

2. *Klebsiella pneumoniae*

Community-level signatures of ecological succession in natural bacterial communities

November 29, 2019

*Alberto Pascual-García^{1, 2, *} and Thomas Bell¹*

(1) Department of Life Sciences. Silwood Park Campus. Imperial College London, Ascot, United Kingdom

(2) Current address: Institute of Integrative Biology. ETH-Zürich, Zürich, Switzerland

() Correspondence: alberto.pascual.garcia@gmail.com.*

RESPONSE TO EDITORIAL COMMENTS

There were not specific editorial comments to respond, other than addressing the specific comments of the reviewers.

RESPONSE TO INDIVIDUAL REVIEWERS

Response to Reviewer 1 Comments for the Authors

In this manuscript, the authors re-analyze a previously published 16S rRNA dataset collected from microbial communities in beech tree holes (a habitat used by this group for some time now as a model for microbial biogeography). The authors take the unorthodox approach of classifying the communities a-priori based on their community composition and then attempting to characterize the resulting 6 community classes according to the results of different physiological assays (also previously published). The second part of the paper is dedicated to predicting the communities' metagenomes using PiCRUST. Noting that the main drivers of beta diversity were sampling location and sampling time, the authors use the PiCRUST results to construct a model of functional community succession in beech tree hole microbial communities.

While the analyses were interesting, this manuscript suffers from some weaknesses. The way it is presented is rather disorganized and the text needs through editing for grammar and for simplicity. The methods are not presented in a way that can be reproduced (only a cursory description of software tools that were used it provided). Furthermore, I have some reservations regarding the use and interpretation of PiCRUST here.

AUTHORS: We thank the reviewer for the positive summary. In the reviewed version, we improved the presentation of the paper emphasizing the assumptions and hypothesis behind the methods used and we restructured both the Main Text and the Suppl. Material to make it more accessible.

Specific comments in order of appearance:

AUTHORS: We have corrected all the typographic errors that have been indicated. Some of the suggestions were no longer relevant in the substantially-modified Introduction.

Abstract: edit: Have ->has

Line 9: edit: major major

Line 10: edit: edaphic

Lines 10-13: the authors provide very little support for this argument. The claims about macroecological results being spurious and difficult to validate are not supported at all by citations. Only the claim about the niche vs neutrality question is supported by a citation but it's not clear to me how this is a problem resulting from the analysis approach, nor how the approaches laid out in this paper will help resolve it.

AUTHORS: We considered the reservations on the Introduction of this and the second reviewer, and we now present an Introduction which more explicitly confronts the caveats of the methodologies typically used in the field. In the sentence mentioned by the reviewer, we were attempting to say that there is a tendency to apply the same methods used in macroscopic systems to microscopic systems and this is not always a good strategy. In the new introduction we removed this reference to macroecology to avoid confusion.

Line 13: extra comma

Line 13-14: Again, I do not disagree with this argument about "context dependence", but it needs to be supported with citation. Same goes for the argument about "fishing expeditions". Could the authors point to an example of such a fishing expedition?

AUTHORS: With fishing expeditions we meant studies measuring many environmental variables looking for correlations between variables and functions. Beyond the fact that increasing the number of measured variables increases the probability of finding a spurious correlation, these variables are themselves strongly correlated (which makes it even more difficult to understand their relative role) and it becomes very difficult to identify which factors impact microbial communities (see e.g. Ref. [1] in the end of this report, and included in the new version of the manuscript).

Line 23: How is this not a fishing expedition? You would still generate "dozens" of measurements to try to identify which explains you're a-priori defined community types.

AUTHORS: We aim to isolate the "composition-function" relationship from the "environment-composition" relationship by growing the communities in a complex environment under laboratory conditions. Although the measurements may depend on the environment selected, it is the same for all the communities and hence differences can be attributed to the composition. The approach would have failed if the community classes we identified did not show functional differences. We believe this approach is quite different from simply searching for correlations in complex datasets. We have tried to improve the language to make this as clear as possible. In addition and following a reviewer's suggestion, we performed an additional randomization in our SEM analysis proving that the relation between community classes and environments is not spurious (see below).

Line 44: Alternatively, the differentiating environmental parameter was not measured

AUTHORS: In this setting, in which the environment remains constant for all communities, we would conclude that the communities are redundant under these conditions. We note that the approach we use does not rely on identifying the environmental parameter responsible for different compositions/functions.

Line 56: "result in functions to"?

Line 58: edit: analysis -> analyses

Line 63: edit: Beech tree

Line 65: the term phytotelmata is introduced here quite suddenly.

AUTHORS: We now define it more carefully in the reviewed version.

Line 67: It is hard to say that "influence on microbial communities is largely unknown" given the large body of work produced by this group on these very communities.

AUTHORS: The work of this group has mainly focused on the relationship between composition and function, while the current work focuses on the relationship between environment and composition, which remains largely

unknown. This work represents one of the first attempts we make to address this question. We believe it is fair to say that these environments are poorly characterised relative to many environments.

Line 85: “similar compositions and functions were translated into different classes of genetic repertoires, susceptible of a functional interpretation.” I do not understand this sentence. What is the different between “functions” and “genetic repertoires” in this context? Is the term “function” used to describe only the results of the physiological/enzymatic assays or also the PiCRUST results?

AUTHORS: We thank the reviewer for this comment because we realized that there was some lack of consistency to differentiate both. Now we use “functions” to describe the physiological/enzymatic assays and the “genetic repertoires” are presented as a mechanistic support of the functional results found.

Line 86: resembling ->resembling. In single cells, do the authors mean mono cultures? In any case, a citation is needed here.

AUTHORS: We were referring to the physiology of single cells, which are mostly studied in monocultures (with some very recent exceptions of groups using microfluidic devices having single-cell resolution). Please see the next comment.

Line 87: I do not think “we believe” is an appropriate term for a scientific article. Is the succession an inference? A hypothesis?

AUTHORS: We rephrased it as follows: “Interestingly, interpreting the signatures found in the functional measurements and in the genetic repertoires led us to hypothesize the existence of community-level ecological strategies reflecting an ecological succession driven by local environmental dynamics of the tree-holes. These ecological strategies resemble the classical distinction between r- and K-strategists described for single species [Andrews and Harris, Springer (1986)].”

Line 90: This would be a good place to refer the reader to table 1 in the supplement and to the unnamed figure below it.

AUTHORS: Done.

Line 93: This referral to Figure 1 does not give sufficient information to convince the reader that there are indeed 6 classes. First, the authors should also refer the reader here to sup. Fig. 3. Second, just by looking at the figure, I cannot be convinced. The lines connecting the points to the cluster centroids are misleading and distract from the fact that there is substantial overlap in the ordination between the groups (the lines also make this figure difficult to read in general, and I’m not sure what purpose they serve, and the numbers in the centroids are barely legible). The methods state that the PAM method was used to determine these clusters and that six clusters provided the optimal CH index. This step is a critical step for all subsequent analyses. Could the authors provide this data somehow?

AUTHORS: We thank the reviewer for this observation that we acknowledge was generating confusion, since also reviewer 2 raised concerns on this figure. Since the main method to find the classes is the PAM method, as pointed out by the reviewer, we removed the PCoA and provided a plot including the CH index against k. In addition, the relationship between the samples’ location and the classes is now represented in the dendrogram (previously a Suppl. Fig.) that we think it is also pertinent to move to the front due to other concerns expressed by this reviewer below.

Also, when you call the figure in the text, are you calling figure 1a or 1b? please be explicit. More on Figure 1: should be community classes not “communities classes”, genera not “genus”, within not “into”, throughout, not “alongthrough”. Also, genus names should be capitalized (and italicized)

AUTHORS: We thank the reviewer for these corrections that we incorporated in a new figure.

Line 94: communities #1 and #3 are dominated by pseudomonas, but this is not mentioned.

AUTHORS: The observation of the reviewer is correct, we corrected the term “dominant” because we aimed to denote the communities according to the most *representative* compositional signature. Although it is true that Pseudomonas is also abundant in classes 1 and 3, representing them at the species level (Suppl. Fig. 9) reveals that classes 4 and 6 are dominated by Pseudomonas putida while 1 and 3 have a more even composition of Pseudomonas species. This is now explained in the text.

Line 98: This would be a good place to call supplementary figures 1 and 4 (generally, the order of the sup. Figures appears to be rather arbitrary).

AUTHORS: We thank the reviewer for the suggestion. Suppl. Fig. 1 is now in Fig. 1 at the Main Text. Please see comment on Line 108-119.

Line 105. Class 1 encompasses the largest number of communities. This seems like an important piece of information. In a traditional analyses, the number of samples between different treatments would have been balanced. How balanced are the different clusters? Could the authors present the number of samples within each cluster and their location in an easily readable fashion?

AUTHORS: The clusters are approximately balanced except for class 1, which has a substantially larger number, and this is why it was chosen as the reference community in the SEM analysis. The influence of an unbalanced design on the results is a fair point, so now we provide additional quality tests for both SEM and PiCRUST (see below). The number of samples are presented at the Suppl. Table 2 in the new version, and two representations of the distribution of the samples are provided: i) the dendrogram in Fig. 1 in the Main Text and Suppl. Fig. 10 shows several examples.

Lines 108-119. I'm familiar with ANOSIM and with distance-decay relationships but I simply could not understand what the authors are doing here. It is not helpful that the methods employed here are partitioned seemingly at random between the Methods and the Supplementary Methods sections, or that no analysis code was provided.

AUTHORS: We apologise for this lack of clarity. The rationale to organize the Methods in this way was that, given its length, we aimed to provide a self-contained Suppl. Material while bringing summarized methods in the Main Text. Since Suppl. Material was split into Suppl. Results and Suppl. Methods, with figures extending results in the former section and methodological figures in the latter, the order of presentation in the Main Text sometimes reflects this organization with jumps in the order of the figures. We now simplified the Methods, making those in the Main Text more self-contained, and reorganized the Suppl. Material to present them in parallel with the Main Text. Regarding the method, we created synthetic data to illustrate the use of this statistical method and its interpretation and, instead of performing the previous randomization, we now combine the analysis with other metrics that allow us to more clearly show our arguments. Finally, we also provide the code for the computation of this and other methods used in the paper, available in https://github.com/apascualgarcia/TreeHoles_descriptive.

How many spatial clusters of samples are included within each cutoff?

AUTHORS: We provide a table with these values in the new Fig. 2.

Is the ANOSIM statistic computed for comparison between spatial clusters within each special cutoff?

AUTHORS: This is correct.

Why is a high R^2 interpreted as high similarity?

AUTHORS: We apologize because the correct terminology for the ANOSIM is R , and not R^2 , this is now corrected. The R ANOSIM statistics computes the relative similarity of samples within clusters with respect to samples between clusters, and it is intrinsic to the method a permutation test to assess the significance, whose results now we provide as Suppl. Figures. Higher is its value larger the within- vs. between-clusters differences of the distances' ranks. A question that we overlooked in the previous version and that it may be the motivation behind the reviewer's question, is that the R value could be equally high for different ratios of the similarity within/between-clusters, because the metric works with ranks. This possibility motivated us to understand better this metric and to compare it with other metrics. We found that the ANOSIM statistics is sensitive to an increase in the variance in the β -diversity distances, implying that the probability that distances between clusters have comparable values to those within clusters increases, hence reducing the ANOSIM value. Together with the MRPP statistics we now show that the mean distances within the clusters of the different classifications created at different orders of magnitude of the spatial distances increases and that, beyond the mean values, there must be a large variance to explain the decay of the ANOSIM statistics. Since the null hypothesis explaining a decay in the similarity of the communities is that the assembly is driven by stochastic processes with some limits for the scope of bacterial dispersal, this result is our starting point to explore the alternative hypothesis stating that local environmental conditions may promote that communities sampled at distant locations are highly similar in their compositions.

Doesn't the $\log(5)$ threshold simply include all samples? How is an ANOSIM statistic computed in this case?

AUTHORS: At this threshold there are still two clusters separated by more than 100km. We show these two distant areas in an illustration in the new Fig. 2.

Why didn't the authors simply plot pairwise dissimilarity against spatial distance and compute a Mantel test between community dissimilarity and spatial distance matrices?

AUTHORS: A Mantel test brings significant results ($r = 0.21; p < 10^{-3}$ for D_{SparCC} and $r = 0.19; p < 10^{-3}$ for D_{JSD}) but it does not tell us how this similarity changes with increasing/decreasing spatial distances. Addressing this question with a Mantel statistics would imply developing a procedure similar in spirit to the one we developed. But, while for ANOSIM the procedure is quite natural because, at each threshold, we can split the data in two (intra- and inter-cluster distances), for Mantel tests it is not clear how this splitting should be performed and combined into a single metric.

Line 115: communities'

AUTHORS: Corrected.

Line 123: "the specific date turned out to be more informative than the site" Isn't this at odds with the dendrogram in Figure S1? It clearly seems as if Site is the strongest driver of community dissimilarity

AUTHORS: We respectfully disagree with the reviewer on this point. We have moved the dendrogram to the Main Text to make this question clear. In the caption, we explain three examples (highlighted with dotted lines), showing that the colours in the bar corresponding to the date are in better correspondence to those of the classes than the bar corresponding to the site. Moreover, beyond subjective inspection of the figure, results in Suppl. Tables 3 and 4 show that the clusters corresponding to the day of sampling are more significantly associated to the classes.

Lines 127-129: Could the authors rephrase this sentence? It is very difficult to read. Also it's not clear what is the part Figure S1 in making this argument.

Line 135: Please edit this text: "there were also measured ATP"

AUTHORS: We modified the text.

Line 131-135: The authors should make sure and be clear about the fact that these experiments were already published elsewhere. Furthermore, isn't this an example of the very same "fishing expedition" that the authors explain in the introduction that their method is designed to avoid? Why select these metrics and not others?

AUTHORS: We made clear in several parts of the article that the experiments were published (in Introduction, Results and Methods). We refer to a fishing expedition as a search for environmental variables to explain one functional measurements (a "many to one" search with microbial communities mediating the relation). The more variables are measured, the higher the probability of finding a spurious correlation. In our work, we have a single environment and we quantify functions that are directly generated (and not mediated) by the communities. This means that any of the scenarios that can be found (functional redundancy or functional differentiation) can be attributed to microbial communities, independently of whether we measure one or many variables. It is true that measuring more variables increases the probability of rejecting the hypothesis of the prevalence of functional redundancy, but this is a causal result (not a statistical one) and there is no risk until we interpret the results via the genetic repertoires (because, again, we move to an analysis of "many" genes to "one" functional signature). Therefore, only after verifying the consistency of the relationship between the genetic repertoires (e.g. sets of genes and not singles genes supporting an observation) and the functional signatures did we report the result as a robust finding.

Another point that is important to make, is that these aren't measurements of the activities of the communities themselves, but rather of communities that have arisen as a result of cryo-preservation, thawing and growth in a lab-made media. Ref. #18 is somewhat vague about this, but if I understand correctly, the communities were sequenced before, and not after the mesocosm experiment. How can we be sure that the 6 community classes are maintained during/after the mesocosm experiment? Perhaps their phylogenetic makeup completely changed?

AUTHORS: The communities were grown for 7 days and then sequenced, isolated and cryo-preserved. The functional assays were performed reviving the communities and growing them in the same media, and the measurements were performed after seven days. We agree with the reviewer that the communities could have changed, and note this caveat in the Methods. However, we further note that, even if the communities did change, we

were able to detect significant effects of the initial community classes, thus demonstrating a strong legacy of these classes for the functional measurements irrespective of the final community compositions.

Line 138: Supplemental figure 5 seems to appear in the text before sup. Fig 4. In fact, the order of the supplemental figures in the text is as follows: 3,2,1,5,24,4,10-12,15,16,17,

AUTHORS: Thank you for pointing this out. We answered to this point in the comment to Lines 108-119.

Line 138: What do the authors mean when they say that functional capacities were distinct between communities? How is this measured, and how does Sup. Figure 5 show this?

AUTHORS: Suppl. Fig 5 (Suppl. Fig. 12 in the reviewed manuscript) illustrates the different functions across classes which, in some cases, are notably different looking at the separation of the densities plotted in the axis. The reviewer is right in that we cannot say anything about the significance simply by looking at these figures, whose role is only to illustrate how the measurements are distributed. We address the question of the significance of the differences between classes in their functional performances using Structural Equation Modelling, finding that some functions are significantly different and others are not, and that it is possible to find distinctive signatures for each community class. We clarified this as follows: “Visual inspection indicated substantial differences in the functional capacities among the community classes. In some cases, communities belonging to different classes clearly separated, as shown in the histograms in Suppl. Fig. 12. Therefore, we explored if these differences among the community classes were significant using structural equation models (SEM)”

Line 147: “That different coefficients for the classes in some pathways brought a better fit than the model with a single parameter for all classes, supported the hypothesis that the classes had differentiated function performances.” I do not see how this constitutes a valid statistical test. Is this outcome significantly different than an outcome generated by reshuffling the labels (creating sets of 6 classes)?

AUTHORS: In our methods we explored a large number of different models and we used several metrics to assess if one model is better than the other. The main criteria to accept a new model is that the AIC is lower, but we also considered a number of other metrics (e.g. Tucker Lewis index), and we should observe a systematic improvement in most of these metrics to accept a new model (they are typically very consistent across the metrics). Nevertheless, we acknowledge the complexity of the procedure, so we now include a table that describes the metrics for the following scenarios, now explicitly explained in the text: i) All parameters are constrained to be equal; ii) all parameters are free of constraints; and iii) model that are intermediate between both situations (the final model belongs to this scenario). In addition, for the final model we shuffled the samples as suggested by the reviewer. Shuffling the samples means that each class has a mix of samples coming from all classes, and hence the estimated pathways should be independent of the community class (i.e. uninformative in distinguishing the classes). We found a dramatic decrease in the metrics assessing the significance of the model using this procedure and, as expected, that the coefficients were similar for all of the community classes (see the new Suppl. Fig. 15), which supports our interpretation that the observed classes are truly indicative of different functional performances.

PiCRUST analysis: PiCRUST is a type of program that should used with caution, because of the underlying assumption that the phylogenetic composition is a good predictor of a genetic repertoire. Certainly, broadly this assumption is true, but there are some important caveats and exceptions to this. Importantly, the results depend on the genome database used. Since the genome database is highly skewed towards human-derived bacterial genomes (and indeed this method was designed and benchmarked with human samples in mind), using this method in environmentally-derived samples could be risky, in particular in a rather exotic environment that has contributed a relatively small amount of genomes to the database. This can be potentially circumvented by (i) sequencing and curating a genome database that is specific to this environment or, more realistically, (ii) sequencing metagenomes from a subset of the 753 samples and comparing the PiCRUST predictions to the actual metagenome content. This dataset seems particularly suited to the latter option, and I think adding this data would be of great benefit. I am not advocating that the authors not use PiCRUST, but rather treat the results with the appropriate amount of caution and attempt to validate the resulting predictions.

AUTHORS: We agree with the concerns of the reviewer. We have documented whether this is likely to be a problem by computing the NSTI score, which evaluates how far the OTUs found in our samples are from those used for the predictions. We found an NSTI score of 0.059 which should give us reasonable confidence of the PiCRUST predictions, as noted by Reviewer 3. The reason for finding low values for the NSTI score is that there are many taxa in this system that have close relatives to gut bacteria. We also computed the NSTI for each class (see response to reviewer 3) and we did not observe a remarkable difference across classes. We agree in that

having the actual metagenome would be of great benefit, however this is not a possibility at this stage of the project, among other reasons because to make results comparable the sequences for the metagenomes should be collected at the same time than those used for 16S rRNA sequencing. Moreover, we believe that the consistency of the data across all the analysis presented is indicative of having a meaningful prediction, but we agree in that we should also acknowledge its limitations. In this respect, we did our best in being cautious in the interpretation of our results.

It seems like one could roughly classify the classes into two groups: classes 1,3,5 and classes 2,4,6. However, while in Figure 1 it is apparent that classes 1,3 and 4,6 are relatively similar, classes 2 and 5 are outliers, dominated by completely different genera (Paenibacillus, Serratia). So, the phylogenetic profiles appear to be disconnected from the functional profiles. Classes 2 and 5 are actually discussed as outliers in the text because class 2 is the sporulation class and class 5 the glycolytic one.

AUTHORS: The distinction between these two groups (classes 1,3,5 and classes 2,4,6) is supported by several observations and we comment it in the manuscript. Nevertheless, there are differences between the classes within each of these two groups such as those pointed out by the reviewer (sporulation or glycolytic activity) that are ecologically meaningful, and that might explain the discrepancies in the phylogenetic profiles. We do not consider these classes “outliers” in the metagenomic repertoires; we have tried to present a scenario in which there is a continuum within each these 2 groups, but also a separation between both groups. These observations motivated us to interpret the results in terms of ecological succession: the first group (1,3,5) being early colonizers and (2,4,6) late colonizers and class 5 and 2 being the first and the last, i.e. a succession $5 > 1 > 3 > 4 > 6 > 2$.

Furthermore, the authors should consider the compositionality of this data, especially when observing changes in ubiquitous machineries such as the TCA cycle. The results could be affected by differences in predicted mean genome sizes, or by the relative abundance of un-annotated genes.

AUTHORS: To mitigate these potential biases, we normalized the metagenome predictions by the number of copies of each gene found in each genome, and we considered the number of un-annotated genes in the tests when quantifying the mean proportion of genes across different sets of samples.

Response to Reviewer 2 Comments for the Authors

Here the authors present a compelling dataset of bacterial communities in beech tree holes, which they use as a model to assess bacterial community clusters and ecological succession. Overall the ideas, data and methods are sound. However, I think for background and substance is needed to elevate the reach of the paper. First, community clusters. This is a great example of the use of community clusters, but more content in the introduction is needed to really help guide the field to using this method more often (I also suggest a comparison to traditional diversity measurement, so show how clustering can be more informative). Second, the beech tree hole system is likely low diversity (it is not clear), regardless, better set up why this is such a good model system and how it can help inform assessments in other more diverse (or less) environments. Finally, the ideas around ecological strategies need to be better explained in the introduction so results and discussion are more understandable. Currently the results are quite long and don't do enough to push the main thesis. Overall, an interesting effort and I look forward to the next round.

AUTHORS: We thank the reviewer for the constructive comments. We have made a substantial modification in the Introduction to clarify our assumptions and hypotheses, and identified how our approach differs from others. We think that the most valuable contribution we make is the fact that we are working with “domesticated” communities grown in a common (and complex) media. This is how we decouple microbial composition from environmental variables, and the reason why a clustering of communities allows us to investigate the relation between composition and function. By contrast, more typical approximation need to relate environmental variables and composition and then composition and function from survey data.

Abstract: - 700 samples, give the geographic reference of where/distance covered. - Define community classes - Split up the sentence beginning with “We used structural equation modelling...” Also, it doesn't quite make since what ‘their’ is referring to. - Metagenomic predictions - Based on what? - Specify the ‘ecological succession process’ that was observed- which taxa and why. - Likewise, the last sentence could be more specific - How can this system inform on other systems? Or at least indicate why this system is a good model to base understanding of other microbial communities

AUTHORS: We reviewed the abstract considering the new changes in the Introduction and the new results. We did our best to clarify the text based on the reviewer comments within the space constraints of the journal.

Introduction - When classifying communities, presumably the diversity of the system will also effect how clusters are built. It would be good background to add in information on this. Human communities tend to be much more simple than soil communities (for example), so will clustering work equally well on high diverse samples?

AUTHORS: This system has a relatively low diversity, and we agree that increasing the number of species will make it more challenging to cluster the communities.

The ability of metagenomic analysis to result in hypotheses is limited by annotation quality – which is still limited. This should be addressed with a reference or 2.

AUTHORS: We agree with the reviewer. In our response to reviewer 1 we explained that the most important species in our system can be observed in human gut, so the annotation is somewhat better than would be expected from a poorly-characterised environmental sample, which we now report in the Main Text. This is reflected in the low NSTI score found (a metric quantifying the quality of the prediction), which we also provide now in Results, and additional analysis in Suppl. Material in response to comments of reviewer 3.

‘To investigate these questions’ be more specific, there were actually a lot of gaps brought up. Same with the following sentence.

AUTHORS: In the reviewed Introduction we link explicitly our assumptions and the pipeline proposed.

I would like to see come comparison/ contrast hypotheses set up- between what has been found with macro/meso organisms and what is expected with the micro community. This would then make the topic even more relevant – could more strongly bring in biogeography, address if microbes are structured by same processes and macrobes... etc.

AUTHORS: Possibly one of the most important challenges in microbial biogeography is to disentangle the relative role of niche and neutral effects in the assembly of the communities. We reconsidered the analysis of the decay of the similarity between the communities, creating synthetic data and using two more metrics, what allowed us to investigate more explicitly this question. We restrain from making a thorough discussion of the relationship between the results found and patterns in macroscopic systems due to the lack of space, and because we believe there is no clear consensus yet about which are the dominant factors operating in microbial communities, with different results found for different systems.

The section about decomposition/function needs a better introduction/background in the previous paragraphs. - ‘we believe’? word choice. Gives evidence maybe.

AUTHORS: We have substantially revised the introduction.

Results - The colors should only be listed in the figure legend – in the text explain the results and statistics.

AUTHORS: We now refer to the classes by a selected OTU, and we removed references to colors and class numbers in the Main Text.

Where are the statistics to go along with the PCA and how much each axis explains?

AUTHORS: The query is similar to one raised by reviewer 1, which we address above. The PCoA was used for illustration purposes. We obtained the community classes using an agglomerative clustering and searching the optimal cut-off. We therefore decided to remove this figure to avoid confusion. We have retained the PCA of the metagenomes to illustrate the orthogonal gene content of the different classes (see Suppl. Results), and we indicate the amount of the variance explained by the PCA axes.

The sentence about uneven sampling and the specific sites. More context is needed, because alone the name of the sites don’t mean much.

AUTHORS: We now bring more context and we improved Fig. 2 to illustrate these areas.

Some basic metrics of the samples would be helpful, how many sequences/sample, otus/sample, and put it into context.

AUTHORS: This information is now in Suppl. Table 2 in the revised manuscript.

Since the argument that this method will progress the field more, can you compare the cluster results to a typical community diversity Mantel/Permanova. But regardless a comparison would help the argument of the paper.

AUTHORS: We now include the values of a Mantel test as well as the mrpp statistics, which reinforces the results found using the ANOSIM statistics. In the Suppl. Material we compared these results with a Permanova test. To illustrate our interpretations, we now provide the results of the different metrics used for synthetic datasets.

“To explore . . . functional performance. . . [18]”

- When referring to the clusters – sometimes it is by the number and sometimes by the dominant taxa. I think it should always be by the dominant taxa, this is more informative to the reader and could then be more relevant for future work using similar methods.

AUTHORS: In the reviewed version we refer to the classes by their representative taxa. As we noted in a response to reviewer 1, we do not necessarily select the dominant taxon (since some dominant taxa are shared among classes), but we select a representative taxon that distinguishes the class.

DISCUSSION - Expand reasons for distance decay finding - There is a lot of speculation about the variability/weak patterns. I would like to see this go beyond just saying ‘it could have been this’ and really address how this study will help move the field forward. There are interesting analyses that are being overlooked for space on the variability. - This would also leave more space to really get into hypotheses to link clusters and function.

AUTHORS: We thank the reviewer for this comment that motivated us, together with a comment from reviewer 1, to improve our analysis generating synthetic data and including new metrics. The new results show how the rejection of the null hypothesis (that the distance decay observed is due to stochastic processes) lead us to investigate the existence of niche effects, which naturally motivates the search for clusters and its relation with functional signatures.

Response to Reviewer 3 Comments for the Authors

Authors examined beech tree-holes bacterial communities at large scale and linked six distinct community classes to community-level functional capacity. It is a well-designed top-down approach to link potential bacterial metabolic functions predicted from simple in situ communities to functional performances. It is interesting to see the community types and functions are linked to ecological succession in a predictable way. The study is unique and the manuscript is also well organized. I have only several minor comments below.

AUTHORS: We thank the reviewer for the positive summary.

Minor comments There are no line numbers, which makes hard to leave comments line by line.

AUTHORS: We apologize for this. The initial manuscript was transferred from biorXiv and the editorial board asked for a new version with line numbers, but apparently only reviewer 1 received that version.

In this kind of simple communities, phytotypes defined at 97% 16S similarity seem too broad. How about clustering sequences at sub-OTU level such as 99% similarity or exact sequence variants ESV? Around 100% identity threshold for the 16S V4 region was suggested as a proxy for bacterial species (Edgar, 2018).

AUTHORS: We agree with the reviewer in that the resolution of the phylotypes considered is an important question. For instance, to compute the metagenomes, the PiCRUST pipeline requires starting with the ESVs found in each sample, because the aim is to retrieve the genetic repertoires as accurately as possible. However, the finer OTUs picking scheme available considers 97% sequence identity. Therefore, we decided for consistency to also use this level of resolution for the OTUs to detect the classes. Despite of these technical limitations, we also believe that the level of resolution of the description we are bringing in this work does not require a finer phylotype, because we are not presenting statements about specific strains (that could perhaps be more relevant for instance in the context of human disease) but rather on coarse-grained classes encompassing tens to hundreds of communities. We therefore retain the 97% threshold while acknowledging that finer taxonomic resolution could further refine the study.

Another concern is that the total bacterial abundance (or biomass) does not seem to be accounted in this community type vs. function comparison. Is there any information at which orders of magnitude microbial abundance vary among samples?

AUTHORS: The number of cells of the community is one of the variables we considered in the structural equation modelling analysis, and so should be taken into account in the analyses.

For example, the relatively high proportion of Paenibacillus in class 2 may not parallel their absolute abundance. What if their absolute abundance remained at similar level across all samples?

AUTHORS: In our case, the absolute abundances can be estimated by multiplying the cell numbers by the relative abundances, in the following table we provide the median of these values for the most important OTUs in each class. *P. borealis* in class 2 doubles the number of cells found in the other classes.

Num. reads	P. borealis	K. pneumoniae	P. putida	S. fonticola
Class 1	10155	50544	15476	34371
Class 2	25931	8264	8123	22038
Class 3	13424	32224	14042	24033
Class 4	10953	4924	5825	4042
Class 5	11855	9749	3779	80690
Class 6	3552	1588	7790	2408

Then you still can explain that their dominance is due to the high content of sporulation genes?

AUTHORS: Please note that our argument around sporulation genes is that, together with the low diversity of these communities and other observations such as the negative relationship between chitinase activity and cell's number (see Discussion) suggests that the environment in which these communities live is low in resources. We can just remain agnostic on whether the dominance of *Paenibacillus* is a consequence of having sporulation genes.

In Methods Dataset 1. Why did you retain >100 reads samples? 100 reads per sample seems too low to be included in any further analysis. Even if you provided in supple that sparCC distance index is not affected by the number of reads, I do not think 100 reads are sufficient enough to reflect real community diversity. Also, those low read numbers are commonly obtained from incomplete PCR reaction or incomplete amplicon sequencing in MiSeq due mainly to low microbial biomass or the presence of PCR inhibitors. Could you provide any evidence of the same or similar results without those too low reads samples?

AUTHORS: We believe there is some misunderstanding here. In Methods we stated that: “We considered only samples with more than 10K reads, and were removed species with less than 100 reads across all samples.” Therefore, the communities have at least 10K reads (we agree with the reviewer in that the number would otherwise be too low), and what we removed were species with too low read numbers.

Operational taxonomic units (OTUs) typo

16S rRNA gene sequence similarity. Typo

AUTHORS: Thank you for the corrections.

In Discussion It is highly speculative that the dominance of Paenibacillus in class 2 is ascribed to the large fraction of sporulation and spore germination genes.

AUTHORS: We agree, and we did not intend to suggest this; we do believe this observation supports the hypothesis that the environment is low in resources.

Different from other class-representative genera, there are not many Paenibacillus genomes available, which reduces the accuracy of 16S-based metagenome prediction as discussed below. Is it supported by any evidence obtained from spore counting assay or endospore enrichment experiment?

AUTHORS: We discuss this question below.

Cells having those genes in their genomes are not necessarily dormant or present in spore forms.

AUTHORS: We agree with these statements. We have read through the text and tried to ensure that we do not imply this direct link. Connecting the PiCRUST data to the compositional data will necessarily be speculative and require further study but the observation is consistent with our observaton.

In Supplementary doc 1. The mean weighted NSTI score of 0.059 seems appropriate to use PiCRUST-base gene prediction. However, functional gene prediction results should be carefully interpreted when the number of neighboring genomes varies across major taxa. The number of available neighboring genomes of those representative

taxa (Serratia, Klebsiella, Pseudomonas, and Paenibacillus) of each community class is different. There are not many genomes available for Paenibacillus compared to those of other three genera and in particular P. borealis has only one genome available.

AUTHORS: To address this question, we computed the NSTI score for each class to see if there are substantial differences across classes. We observed that most of the confidence intervals overlap, having classes 4 and 5 significantly lower values (meaning better predictions) than the rest of the classes (Bonferroni-corrected Wilcoxon test). However, we do not observe any class with significantly worst NSTI scores than the NSTI score value observed from the whole dataset which, as the reviewer pointed out, is appropriate.

NSTI score	LowCI	Median	HighCI
Class 1	0.060	0.063	0.067
Class 2	0.058	0.062	0.070
Class 3	0.056	0.060	0.064
Class 4	0.047	0.051	0.058
Class 5	0.045	0.049	0.055
Class 6	0.055	0.061	0.067

We would like to thank again the Reviewers and the Editor for the critical comments that we believe substantially improved our manuscript.

References

- [1] Emily B Graham, Joseph E Knelman, Andreas Schindlbacher, Steven Siciliano, Marc Breulmann, Anthony Yannarell, JM Beman, Guy Abell, Laurent Philippot, James Prosser, et al. Microbes as engines of ecosystem function: when does community structure enhance predictions of ecosystem processes? *Frontiers in microbiology*, 7:214, 2016.

Reviewers' comments:

Reviewer #1 (Remarks to the Author):

I apologize for the length of the time it has taken me to review this manuscript. This version is significantly improved in comparison to the first version, and its topic and analytical approaches remain interesting.

However, I found the text highly inaccessible, at times I could not follow the authors' logic and I think that the conclusions are not always supported by the data. Importantly, the authors finally conclude that the differences that they observe are due to sampling at different successional stages and that the main driver of this succession is the allogenic factor of timing of rain events. It seems to me that this hypothesis could be addressed experimentally with the experimental system at hand.

I will present the rest of my comments in order. Please note that I marked major comments with the word "Major".

Line 56 interests \diamond interest

Line 58 "this procedure" -> "the above pipeline"

Line 61 emphasis HAS BEEN made ON ...

Line 62 tree holes -> tree hole

Line 63 may help us understand (delete "to")

Line 65 This sentence is rather hard to understand, and I think it is key.

Major: Line 86 As I commented in the previous version, I find it odd that all of the communities were only sequenced AFTER a period of growth in the lab and that the original samples were never analyzed. How can we know what was the effect of growth in the lab media?

Line 99 This is now much clearer to me than it was in the 1st version, and seems like a neat approach, utilizing the different spatial scales in the sampling scheme. One last nitpick: would it make sense to see, as a control, if distance threshold generated from randomized datasets (e.g. shuffling the distances, not the simulated dataset) would produce a similar decay in the R statistic? Supp. Figure 7 needs a more detailed legend.

Line 110 "this result" refers to the actual result, not the simulated data, right? Wording is confusing.

Major: Line 110 Why does a distance-decay in similarity imply high levels of dispersal? On the contrary. A distance decay in community similarity implies there is a dispersal limitation.

Line 115 genus -> genera

Line 127 I'm sorry but I don't see how the previous analysis shows this.

Line 142 this sentence can be made much simpler

Major: Line 145. "The sampled communities were cryo-preserved after sequencing and later revived". In line 88 you state that "communities were grown in a medium... and then .. sequencing". So if I understand the sequence of events correctly, the samples were collected, preserved, revived in media, sequenced, preserved again, and revived a second time. Is this correct? If that is the case, the 16S data and the physiological data are collected from communities at different states, as we have no idea how the cryopreservation and subsequent revival of the communities affect their composition. This seems doubly meaningful when considering the main conclusion from this data is that a temporal factor – succession, is driving these differences.

Line 150 visual inspection of what?

Major: Line 152-162 I'd be lying if I say I understand this. It is not clear to me why this results supports the hypothesis that the classes are significantly different from each other. The authors are saying that a model assuming no distinction between classes had an excellent fit and that a model with some constraints was better. How much better, whether the difference in fit between the models is significant and what this means, I can't say. I'm not sure what is the need to perform these SEMs rather than more standard statistical tests. For the very least the authors can do both, and certainly this needs to be explained in a more clear way.

Line 186. The fact that the same genera are found in these two very different habitats does not

bode well to the predictability of tools like PiCRUST, in my opinion.

Line 187 The PREDICTED fraction

Major: Lines 187-189 I see a some potentially circular logic here. Is the fractions of exoenzymatic genes higher in Paenibacillus (etc) because the Paenibacillus (etc) reference genomes themselves have a high number of exoenzymes? I think it is important to note what is the phylogenetic origin of this enrichment.

Line 197 proteins genes

Lines 202-203 This statement requires a citation.

Line 231 Why only cite these 2 papers there are many more papers that tackle this question, and why focus on soil, this is not a soil environment?

Major: Line 232 Again, I don't understand this logic. You just told us the communities are display spatial autocorrelation /distance decay in similarity, meaning that the more distant communities are from each other the more different they are. You then move on to explain the exact opposite phenomenon – that distant communities are similar to each other – by saying that similar environmental conditions occur at distant locations. This does not make any sense to me.

Line 236 what is meant by "tightly entangled"?

Lines 236-238 this is a repetition of the previous paragraph

Lines 242-243 again, this is an almost verbatim repetition of line 231

Line 244 it is hard for me based on my understanding of the evidence presented to concur that the evidence actually confirms this statement.

Major: Line 256 I think the authors are saying is that the different classes are a result of sampling at different successional stages. If that is the case, I think this point should be made more explicitly. Second, why would the succession not continue once the communities are revived in the lab? Wouldn't you expect that if the communities were allowed to continue growing for longer that the "early" classes would eventually become "late" classes?

Line 320 missing citation

Reviewer #3 (Remarks to the Author):

Authors explained well about my minor comments and the revised manuscript was significantly improved.

Community-level signatures of ecological succession in natural bacterial communities

January 31, 2020

Alberto Pascual-García^{1, 2,*} and Thomas Bell¹

(1) Department of Life Sciences, Silwood Park Campus, Imperial College London, Ascot, United Kingdom

(2) Current address: Institute of Integrative Biology, ETH-Zürich, Zürich, Switzerland

(*) Correspondence: alberto.pascual.garcia@gmail.com.

RESPONSE TO EDITORIAL COMMENTS

There were not specific editorial comments to respond, other than addressing the specific comments of the reviewers.

RESPONSE TO INDIVIDUAL REVIEWERS

Response to Reviewer 1 Comments for the Authors

Reviewer #1 (Remarks to the Author):

I apologize for the length of the time it has taken me to review this manuscript.

AUTHOR: We thank the reviewer for taking the time to thoroughly review our manuscript for the second time.

This version is significantly improved in comparison to the first version, and its topic and analytical approaches remain interesting. However, I found the text highly inaccessible, at times I could not follow the authors' logic and I think that the conclusions are not always supported by the data. Importantly, the authors finally conclude that the differences that they observe are due to sampling at different successional stages and that the main driver of this succession is the allogenic factor of timing of rain events. It seems to me that this hypothesis could be addressed experimentally with the experimental system at hand. I will present the rest of my comments in order. Please note that I marked major comments with the word "Major".

AUTHORS: In this paper we focus on the relationship between composition ("structure") and function rather than the relationship between environmental conditions and composition. We agree with the reviewer that the discussion concludes with a hypothesis suggesting how environmental conditions might have generated the communities we observed, but this is a hypothesis developed in the Discussion rather than being a primary result of the study. We agree this could be tested experimentally, and hope to do so in the future.

Our primary goal (for which we believe we have strong experimental evidence) is that the compositional classes that we observed have distinctive functions, that can be interpreted via metagenomic predictions. The fact that these classes retain information from the location and dates in which they were sampled was not predicted a priori (since the communities were grown in the lab media, as the reviewer points out below) and we took this observation as an opportunity to discuss the ecological processes and build hypotheses for future work.

Line 56 interests -> interest

Line 58 “this procedure” -> “the above pipeline”

Line 61 emphasis HAS BEEN made ON ...

Line 62 tree holes -> tree hole

Line 63 may help us understand (delete “to”)

AUTHORS: We thank the reviewer for the above corrections

Line 65 This sentence is rather hard to understand, and I think it is key.

AUTHORS: We have re-worded the sentence to briefly summarise the main result. Further details of this result are given in the cited reference.

“Previous work using this dataset showed that rare taxa influenced narrow functions (degradation of specific substrates) whereas abundant taxa influenced broad functions (overall community productivity) [32]”

Major: Line 86 As I commented in the previous version, I find it odd that all of the communities were only sequenced AFTER a period of growth in the lab and that the original samples were never analyzed. How can we know what was the effect of growth in the lab media?

AUTHORS: The rationale for doing it this way is that we were primarily interested in the relationship between structure and function rather than between structure and environment (see response to first comment, above). Finding causal relationships between structure and function is made possible here by ensuring that each community is placed in exactly the same environment, as explained in Ref 32. While placing all the communities in a standardised environment creates some problems for interpreting the role of environmental conditions in creating those community compositions, it does allow us to understand how composition relates to functioning since initial environmental conditions are standardised across all 753 communities. Using this approach, the appropriate point at which to sequence the communities is after they have first been passaged on to the laboratory medium since it is those communities that are impacting the measured functions. We have added a new paragraph to the Methods to make this point explicitly.

Despite using this approach, we still observed a strong environmental signal in the sequenced communities (i.e. even after freezing and re-growing under lab conditions), which we thought required some explanation and discussion. We have clarified this point in the results:

“We found that there was a strong relationship between spatial distances and the two β -diversity distances (Mantel test: $r = 0.21$; $p < 10^{-3}$ for D_{SparCC} and $r = 0.19$; $p < 10^{-3}$ for D_{JSD}). This correlation was unexpected because the communities were sequenced following cryo-preservation and subsequent growth under laboratory conditions, so the communities did not necessarily reflect their composition in the original environments.”

Line 99 This is now much clearer to me than it was in the 1st version, and seems like a neat approach, utilizing the different spatial scales in the sampling scheme. One last nitpick: would it make sense to see, as a control, if distance threshold generated from randomized datasets (e.g shuffling the distances, not the simulated dataset) would produce a similar decay in the R statistic?

AUTHORS: We agree with this suggestion, and this approach is indeed incorporated in the evaluation of the significance of the different tests. In particular, the permutation tests performed to estimate the ANOSIM p-values first shuffle the distance classes. Results of this analysis for ANOSIM were reported in Suppl. Fig. 1.

Supp. Figure 7 needs a more detailed legend.

AUTHORS: We improved the legend that now reads: “**Overlap in community dissimilarity in artificially generated β -diversity matrices.** Each matrix has nested levels with different mean distances, α . The overlap of each level with respect to the level with the lowest mean (labeled 0) was estimated as $P(d_{ij}^\alpha \leq d_{ij}^0)$ with d being the β -diversity distance. Standard deviations around the means were $\sigma = 0.1$ (left) and $\sigma = 1$ (right).”

Line 110 “this result” refers to the actual result, not the simulated data, right? Wording is confusing.

AUTHORS: Please see next comment.

Major: Line 110 Why does a distance-decay in similarity imply high levels of dispersal? On the contrary. A distance decay in community similarity implies there is a dispersal limitation.

AUTHORS: We rephrased line 110 and the previous paragraph to clarify this question. Although it is true that the β -diversity similarity decays with the distance and that this pattern could be interpreted in terms of stochastic assembly and dispersal limitation, we suggest that the plausibility of this scenario should be measured against an appropriate null model. We used the ANOSIM statistic to address this question since this method worked well with the simulated data. In particular, we verified that the method is able to capture the spatial scale at which the communities were random. For example, would the communities be very similar at distances ~ 5 m and decay up to e.g. 100m after which they are completely random. In that case, the ANOSIM statistics would be non-significant for distances >100 m.

For real data we observed that, while there is an overall signal of dispersal limitation, (i.e. β -diversity similarity indeed decays with physical distance), the ANOSIM statistics was still significant even when we cluster communities 100km apart. This means that explaining this similarity in terms of purely stochastic processes implies having a sufficiently high dispersal between very distant locations. We think that this possibility is unrealistic, what led us to evaluate the alternative hypothesis: namely that similar conditions at distant locations occur.

Line 115 genus -> genera

AUTHORS: Done, thanks.

Line 127 I'm sorry but I don't see how the previous analysis shows this.

AUTHORS: We hope that with the above clarification we made it clear. We also added this sentence: "This can be noted in the dendrogram of Fig. 2C, which shows how distant sites (dendrogram) could have similar compositions (colour, representing the classes; see also examples in Suppl. Fig. 10)"

Line 142 this sentence can be made much simpler

AUTHORS: The phrase was indeed too long, we splitted it in two.

Major: Line 145. "The sampled communities were cryo-preserved after sequencing and later revived". In line 88 you state that "communities were grown in a medium... and then .. sequencing". So if I understand the sequence of events correctly, the samples were collected, preserved, revived in media, sequenced, preserved again, and revived a second time. Is this correct? If that is the case, the 16S data and the physiological data are collected from communities at different states, as we have no idea how the cryopreservation and subsequent revival of the communities affect their composition. This seems doubly meaningful when considering the main conclusion from this data is that a temporal facto – succession, is driving these differences.

AUTHORS: We characterised how cryopreservation impacted composition in the Rivett et al. (2018) article that we cite (see Supp. Figure 3 in Rivett), which showed a good correlation between composition before- and after cryopreservation, with the abundance of some OTUs being impacted as would be expected.

We stress that, regardless of the impact of cryopreservation on the communities, we believe this comes back to the point we make above: we are here primarily characterising the structure-function relationship rather than a structure-environment relationship. We do note the strong structure-environment signal that we observe *despite* growing the communities under standardised laboratory conditions, which we take as evidence of a particularly strong environmental signal. If we had used the alternative method of measuring functioning in situ, the functional measurements would have been confounded by the environmental conditions, making it very difficult to draw any conclusions about the relationship between structure-function.

As indicated in response to comments above, we have tried to clarify this point in the text.

Line 150 visual inspection of what?

AUTHORS: Thanks for pointing this out. We rephrased it as: "Visual inspection of the functional measurements shown in Suppl. Fig. 12 indicated substantial differences in the functional capacities among the community classes."

Major: Line 152-162 I'd be lying if I say I understand this. It is not clear to me why this results supports the hypothesis that the classes are significantly different from each other. The authors are saying that a model assuming no distinction between classes had an excellent fit and that a model with some constrains was better. How much better, whether the different in fit between the models is significant and what this means, I can't say. I'm not sure what is the need to perform these SEMs rather than more standard statistical test. For the very least the authors can do both, and certainly this needs to be explained in a more clear way.

AUTHORS: We apologize for the lack of clarity. Please note that we used both SEM and more standard statistical tests, which are summarized in Fig. 3D. The reason why we started investigating a SEM is to identify which standard statistical tests would be meaningful. We have 7 functional measurements plus a factor describing the classes, which implies an enormous number of possible models: we do not know which variables are endogenous and which are exogenous, which are the relevant interactions, etc. The purpose of the SEM was therefore to reduce the complexity of statistical test that were employed. Once we had identified the best SEM, we could then explore the role of the community classes. Furthermore, once the model was identified, we could conduct a causal analysis because it was possible to identify the confounding variables for a given endogenous-exogeneous pair of variables analyzed. This allowed us to identify “more standard” models that controlled for the confounding variables while introducing the classes as factors, and allowing us to identify the 8 functional “signatures”.

In summary, we tried to simplify this complex problem into 3 steps. The first step investigated the relationship between the variables, the second investigated whether classes have distinctive functional performances, and the third step used standard models to identify causal relationships between the variables. We looked for optimal models within each step. We note that the two models that the reviewer suggest belong to two different steps in this process.

It could be argued that the SEMs in the first and second step should be compared. We would agree that could be an avenue to explore if, for example, we wanted to identify some sort of optimal partitioning of the functions into a number of unknown groups that maximize the SEM fit. However, in our case the classes were independently identified. We were simply asking whether the the functional performance differs among the classes given a set of classes that are independently derived. We rewrote this section to explain the methodology more clearly.

Line 186. The fact that the same genera are found in these two very different habitats does not bode well to the predictability of tools like PiCRUST, in my opinion.

AUTHORS: The Nearest Sequence Taxon Index is the most important criteria to assess the quality of the predictions according to the developers of PiCRUST, and is reported in the text.

Line 187 The PREDICTED fraction

AUTHORS: The lines immediately above (183-186) explain that we are working with predicted metagenomes.

Major: Lines 187-189 I see a some potentially circular logic here. Is the fractions of exoenzymatic genes higher in Paenibacillus (etc) because the Paenibacillus (etc) reference genomes themselves have a high number of exoenzymes? I think it is important to note what is the phylogenetic origin of this enrichment.

AUTHORS: For communities dominated by Paenibacillus, the fraction of exoenzymatic genes would be higher if the Paenibacillus reference genomes have a higher number of exoenzyme genes- at least that is our understanding of how the PiCRUST algorithm works to predict metagenomes. It was unclear to us why this was circular logic- the predicted metagenomes would not necessarily predict the actual (measured) exoenzymatic rates, nor would the independently derived community classes necessarily reflect different metagenomic profiles.

Line 197 protein genes

AUTHORS: Corrected.

Lines 202-203 This statement requires a citation.

AUTHORS: We included a citation.

Line 231 Why only cite these 2 papers there are many more papers that tackle this question, and why focus on soil, this is not a soil environment?

AUTHORS: We sampled communities within the stirred sediment deposited at the bottom of the tree-hole. We are not aware of autocorrelation results in our system, and hence a soil examples seems most relevant. We agree there is a larger literature- unfortunately, journal regulations limit the number of possible citations.

Major: Line 232 Again, I don't understand this logic. You just told us the communities are display spatial autocorrelation /distance decay in similarity, meaning that the more distant communities are from each other the more different they are. You then move on to explain the exact opposite phenomenon – that distant communities are similar to each other – by saying that similar environmental conditions occur at distant locations. This does not make any sense to me.

AUTHORS: We believe we address this in response to one of the comments above. In brief, it is possible to have a significant but weak spatial autocorrelation that could either be explained by (we believe) unrealistically high levels of dispersal or by local selection.

Line 236 what is meant by “tightly entangled”?

AUTHORS: It means that a factor describing the date of collection and another one describing the sample’s location highly overlap. We now simply say that they are tightly correlated.

Lines 236-238 this is a repetition of the previous paragraph

AUTHORS: We removed this phrase.

Lines 242-243 again, this is an almost verbatim repetition of line 231 Line 244 it is hard for me based on my understanding of the evidence presented to concur that the evidence actually confirms this statement.

AUTHORS: We removed this phrase.

Major: Line 256 I think the authors are saying is that the different classes are a result of sampling at different successional stages. If that is the case, I think this point should be made more explicitly. Second, why would the succession not continue once the communities are revived in the lab? Wouldn’t you expect that if the communities were allowed to continue growing for longer that the “early” classes would eventually become “late” classes?

AUTHORS: This is correct, we explicitly state this point in the previous paragraph, at line 256 starts a second paragraph continuing the explanation. To observe the ecological succession under laboratory conditions, the “early” classes should already contain members of the “late” class, because it is not an open environment as it is in the tree-holes, and we haven’t addressed the importance of local immigration in this work. In addition, it is difficult to reproduce all the abiotic transformations undergoing in the tree-holes. Although the substrate used in the laboratory is complex, it is still far from being as complex as the natural conditions, question such as oxygen variations in interstitial water under drought conditions would be hardly reproducible.

Line 320 missing citation

AUTHORS: We included the citation.

Response to Reviewer 3 Comments for the Authors

Reviewer #3 (Remarks to the Author):

Authors explained well about my minor comments and the revised manuscript was significantly improved.

AUTHORS: We would like to acknowledge again both reviewers for taking the time to review our manuscript

REVIEWERS' COMMENTS:

Reviewer #1 (Remarks to the Author):

I appreciate your patient and serious consideration of my comments and your explanation of the points that were difficult for me to understand. To be honest, I remain slightly skeptical of applying PiCRUST using a mostly HMP-derived database to this tree hole community, but since this is a published and commonly used method, I see no use in stressing this point further. I have no doubt that in the future genomic and metagenomic datasets coming out of this system will resolve this question one way or another.

Answer to the reviewer:

Reviewer #1 (Remarks to the Author):

I appreciate your patient and serious consideration of my comments and your explanation of the points that were difficult for me to understand. To be honest, I remain slightly skeptical of applying PiCRUST using a mostly HMP-derived database to this tree hole community, but since this is a published and commonly used method, I see no use in stressing this point further. I have no doubt that in the future genomic and metagenomic datasets coming out of this system will resolve this question one way or another.

AUTHORS: We thank the reviewer for the insightful comments which greatly improved our manuscript. As the reviewer says, we hope to generate metagenomes in this system in the future, though profiling hundreds of communities is still beyond the resources of our standard funding. For now, we can simply report the quality of our prediction (as measured by the recommended NSTI score) is only marginally higher than mean reported in the original PiCRUST article for some human samples (mean 0.03 ± 0.02), and much better than the mean of mammal samples (0.14 ± 0.06), so we are confident in that the prediction is meaningful. Please see Fig. 3 in the reference below, the median of our NSTI values (0.059) perfectly lie within those of Human samples.

Langille, Morgan GI, et al. "Predictive functional profiling of microbial communities using 16S rRNA marker gene sequences." *Nature biotechnology* 31.9 (2013): 814.)